# Fundamental Problems With Model Editing: How Should Rational Belief Revision Work in LLMs?

**Peter Hase**
*Department of Computer Science*
*University of North Carolina at Chapel Hill*

**Thomas Hofweber**
*Department of Philosophy*
*University of North Carolina at Chapel Hill*

**Xiang Zhou**
*Department of Computer Science*
*University of North Carolina at Chapel Hill*

**Elias Stengel-Eskin**
*Department of Computer Science*
*University of North Carolina at Chapel Hill*

**Mohit Bansal**
*Department of Computer Science*
*University of North Carolina at Chapel Hill*

**Reviewed on OpenReview:** *https://openreview.net/forum?id=LRf19n5Ly3*

## Abstract

The model editing problem concerns how language models should learn new facts about the world over time. While empirical research on model editing has drawn widespread attention, the conceptual foundations of model editing remain shaky – perhaps unsurprisingly, since model editing is essentially *belief revision*, a storied problem in philosophy that has eluded succinct solutions for decades. Model editing nonetheless demands a solution, since we need to be able to control knowledge within language models. With this goal in mind, this paper critiques the standard formulation of the model editing problem and proposes a formal testbed for model editing research. We first describe 13 open problems with model editing, based on challenges with (1) defining the problem, (2) developing benchmarks, and (3) assuming LLMs have editable beliefs in the first place. Many of the challenges are extremely difficult to address, e.g. determining far-reaching consequences of edits, labeling probabilistic entailments between facts, and updating beliefs of agent simulators. Next, we introduce a semi-synthetic dataset for model editing based on Wikidata, where we can evaluate edits against labels given by an idealized Bayesian agent. This enables us to say exactly how belief revision in language models falls short of a desirable epistemic standard. We encourage further research exploring settings where such a gold standard can be compared against.[1]

---

Corresponding author: Peter Hase «peter@cs.unc.edu»
[1]Code for all experiments is provided in the supplement.

# 1   Introduction

Model editing in NLP is an increasingly popular area of research focused on updating the outputs of language models to more accurately reflect the state of the world. Following initial studies (De Cao et al., 2021; Dai et al., 2021; Mitchell et al., 2021; Hase et al., 2021), at least 17 papers were published on the problem in 2023 alone (Betz & Richardson, 2023; Pinter & Elhadad, 2023; Hernandez et al., 2023; Hoelscher-Obermaier et al., 2023; Patil et al., 2023; Yao et al., 2023; Zhong et al., 2023; Han et al., 2023; Hartvigsen et al., 2023; Wu et al., 2023; Wang et al., 2023a;b; Wei et al., 2023; Gupta et al., 2023; Brown et al., 2023; Onoe et al., 2023; Li et al., 2023). Applications of model editing have focused on updating models with changing information over time, unlearning sensitive information, and fixing individual factual mistakes. Indeed, model editing methods now seem necessary given that interventions in the pretraining or finetuning stages of LLM development appear insufficient for solving these problems efficiently (Lazaridou et al., 2021; Dhingra et al., 2022; Debenedetti et al., 2023; Casper et al., 2023a).

Yet the model editing problem stands on shaky theoretical ground. The principal reason for this is that the model editing problem has been framed as an instance of the belief revision problem in philosophy (Hansson, 2022). Past work posits that model editing shares core goals with belief revision, arguing that LLMs should maintain logically consistent outputs when updated with new information (De Cao et al., 2021; Mitchell et al., 2022; Meng et al., 2022a). This means that model editing inherits a host of longstanding challenges regarding how to rationally respond to new information about the world. For example, sometimes new information points to several possible states of the world but is not decisive between them, and determining which of these possible worlds is most likely to be the true world is an unsolved problem (Lewis, 1979).

In this paper, we critique the predominant formulation of the model editing problem and propose a semi-synthetic setting for evaluating model editing with more formality. Our critique of model editing is presented as **13 open challenges, summarized in Table 1** and organized into three categories: (1) challenges with defining the model editing problem, (2) challenges with developing benchmarks, and (3) challenges with assuming LLMs have editable beliefs. On (1), we describe conceptual problems with determining the desired behavior of an LLM after updating on a new fact, focusing on problems of underspecification and unclear goals. On (2), we point out hard-to-overcome issues with developing benchmarks, such as labeling probabilistic factual entailments and constructing datasets for error correction in LLMs. On (3), we suggest that current LLMs may not always have editable beliefs to begin with, and there are problems with manipulating the credences associated with beliefs. Together, these problems demand thorough treatment before model editing work will be able to yield LLMs that can maintain consistent knowledge about the world over time.

In order to provide a cleaner starting point for model editing, we introduce a semi-synthetic setting for evaluating model editing that precisely formalizes the problem, albeit at the cost of tackling a simplified problem with models that are trained from scratch. The key idea of our benchmark is to compare an LLM against a Bayesian model, reflecting that Bayesian epistemology is the gold standard in belief revision (Lin, 2024). Our evaluation uses facts from Wikidata (Vrandečić & Krötzsch, 2014), used to generate a corpus of noisy sentences, which we then train an autoregressive Transformer on. By fitting a Bayesian model to the same data, we are able to obtain exact Bayesian posteriors that serve as the targets we evaluate our language models against. Specifically, our Bayesian model responds to new edit requests, yielding posterior beliefs that we compare our language model against *after model editing* (example test case shown in Fig. 2).

Our experiments show that edits to language models generalize poorly with respect to other relevant beliefs, yielding inconsistent model beliefs. This result is known for pretrained models in certain settings, as measured by assessing model textual outputs (Zhong et al., 2023; Cohen et al., 2024); we further show that language model probabilities consistently diverge from Bayesian posterior probabilities under more general measures of probabilistic coherence. This result helps set the stage for more formal future work on model editing methods.

In summary, this paper's contributions are:

1. A critique of the model editing problem, with a focus on (1) conceptual challenges with its formulation, (2) difficulties with developing benchmarks for the problem, and (3) issues with assuming LLMs possess beliefs about the world and that we can edit those beliefs.

Table 1: Summary of thirteen open problems for model editing with LLMs.

| Open Challenges | Example |
|---|---|
| **Defining the Model Editing Problem** | |
| 1. Problem of Background Beliefs (Sec. 3.1) | • The Raven Paradox: Rational conclusions depend on prior beliefs about the world, which are specific to the LLM being edited (and differ from humans). |
| 2. Problem of Many Possible Worlds (Sec. 3.2) | • If Obama and Trudeau were compatriots, what country would they be from? An edit could lead to many possible worlds. |
| 3. Problem of Complete Corrigibility (Sec. 3.3) | • LLMs should be editable in any way we want, but some edits have many unknown consequences. What if the moon were actually made of cheese? |
| 4. Problem of Missing Context (Sec. 3.4) | • An update like "The Space Needle is in London" is missing context, e.g. a known source, accompanying evidence, or broader conversational context. |
| 5. Problem of Coherence At All Cost (Sec. 3.5) | • How much compute should be used to maintain coherent beliefs vs. achieve goals in an environment? Agentic LLMs face opportunity costs. |
| **Developing Benchmarks** | |
| 6. Factual Entailment Is Hard to Annotate (Sec. 4.1) | • If we learn an animal is a vertebrate rather than invertebrate, we should update our credence that it is venomous. But it is hard to say by how much. |
| 7. Vague and Ambiguous Factual Claims (Sec. 4.2) | • Many common factual claims like "half of Americans are living paycheck to paycheck" (used for model editing by past work) are highly imprecise. |
| 8. Error Correction Requires Targeted, Model-Dependent Testing Strategies (Sec. 4.3) | • To be useful for improving error correction, benchmarks need to specify what errors we want to fix and which models we expect to have those errors. |
| **Assuming LLMs Have Editable Beliefs** | |
| 9. Are LLMs Like Agents or Agent Simulators? (Sec. 5.1) | • Do LLMs have a single set of beliefs, or do they express beliefs of different agents depending on context? Which are we updating? |
| 10. Are LLMs Like Agents or Databases? (Sec. 5.2) | • Do LLMs maintain consistent beliefs or act as passive vessels for data that is curated by humans? |
| 11. No Learned Belief Update Mechanism (Sec. 5.3) | • Why would a small amount of supervised fine-tuning on input-output sequences correspond to an existing, learned belief revision process in an LLM? |
| 12. Not Clear How To Edit Credences (Sec. 5.4) | • LLMs can express uncertainty in language or in next-token probabilities. Which mechanism should be exploited during editing? |
| 13. No Guarantee of Sufficient Memory (Sec. 5.4) | • LLMs have limited parametric memory for storing information. When does learning a new fact demand forgetting something else? |

2. An empirical study of model editing in a semi-synthetic setting with exact Bayesian posteriors for evaluating editing methods against. Our results precisely quantify the degree to which model edits fail to generalize properly and help set the stage for future work on editing methods.

## 2   Background

**Belief Revision.** In philosophy, belief revision concerns how a rational agent should incorporate new information into their existing beliefs. Here, a belief is represented by a sentence in a formal language,

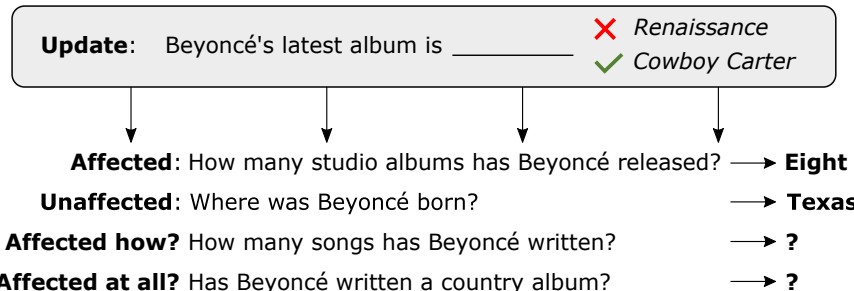

Figure 1: In the predominant formulation of model editing, an LLM's weights are updated so that it gives a new output for a specific input. Even for a simple new fact about the world, however, it can be hard to specify its exact consequences in theory (Sec. 3), or it may be challenging to crowdsource labels for the data in practice (Sec. 4). It is also not clear that LLMs have coherent, revisable beliefs to begin with (Sec. 5).

and agents are typically assumed to be logically omniscient. In the classical AGM theory of belief revision (Alchourrón et al., 1985), an agent aims to maintain logically consistent beliefs in the face of new information. The easiest kind of information to incorporate has nothing to do with previous beliefs; a new belief may be freely formed with no consequences for other beliefs. However, when new information contradicts old beliefs, an agent should "give up beliefs that have as little explanatory power and overall informational value as possible" (Hansson, 2022). There have been many attempts to specify standards for deciding which beliefs to give up, but no single accepted theory has arisen. Importantly, AGM-like theories handle only "full belief" – either something is believed or not. Over time, Bayesian theories have been developed to handle degrees of belief (Lin, 2024). In this paper, we hold Bayesian epistemology as a gold standard for belief revision, but rational belief revision is still not always straightforward in Bayesian theories. In Sec. 3, we describe how known challenges in belief revision apply to the model editing problem.

**Model Editing.** To date, work in model editing has focused on facts of the form $(s, r, o)$, where $s$ represents a subject entity (e.g. *Beyoncé*), $r$ is a relation (e.g. *latest album*), and $o$ is an object (e.g. *Cowboy Carter*). The tuple $(s, r, o)$ represents a factual assertion about the world. The goal of model editing is to update a model with such a fact.

What model editing does in practice is change a model output for a prompt $P$ from an undesired output $o_{old}$ to a desired output $o_{new}$, where the prompt $P$ corresponds to a fact $(s, r, o)$. So, $P$ might be "Beyoncé's latest album is" or "What is Beyoncé's latest album?" See Fig. 1 for an example. Usually, this change is achieved by updating the model weights, for instance by finetuning. Beyond this simple change to one model output, the goals of model editing have been defined in terms of generalization and overgeneralization: we want the model to give the right outputs for datapoints besides the prompt $P$, particularly regarding other potentially relevant facts (also shown in Fig. 1). Past work has categorized these other facts as "entailed" or "neutral" (Hase et al., 2021), as well as "in-scope" or "out-of-scope" (Mitchell et al., 2022). Model outputs are supposed to change in particular ways for these points, generalizing when needed while avoiding overgeneralizing. More broadly, past work has argued that the goal of model editing is to update LLMs with new knowledge while maintaining logical consistency of the model's knowledge (De Cao et al., 2021; Mitchell et al., 2022; Meng et al., 2022a). In this way, the model editing problem shares the same goals as belief revision, and so it naturally inherits the challenges of belief revision, which we elaborate on next.

## 3 Challenges With Defining the Model Editing Problem

This section addresses challenges with defining the model editing problem: what behavior *should* an LLM exhibit after we "edit" the knowledge in a model? We describe how several known challenges in belief revision can make it difficult to answer this question for model editing. Our discussion adopts a Bayesian perspective, as this is the most widely accepted standard for belief revision. The standard argument for Bayesianism is that Bayesian agents can systematically outperform other agents in prediction games that assess the agent's beliefs against reality, like betting on future events (cf. Dutch book arguments; Lin, 2024). Note that we do not

rehash standard critiques of Bayesianism, such as computational intractability, but instead aim to highlight conceptual challenges with specifying ideal behaviors of LLMs after applying a model editing method.

### 3.1 Problem of Background Beliefs

The problem of background beliefs is well expressed in an infamous puzzle known as the Raven Paradox (Hempel, 1945). The paradox goes as follows: suppose you are trying to determine whether all ravens are black. You reason that the statement "all ravens are black" is logically equivalent to the statement "all non-black things are non-ravens" (by contrapositive). Thus, as you go around the world observing things that are not black and not ravens, you increase your confidence that all ravens are black. The paradox is that it certainly seems odd to go through the world without seeing any ravens and end up changing your beliefs about what color ravens are.

It happens that the paradox is resolved by setting a prior on the number of objects in the world (a background belief), which enables one to say how much evidence an observation provides for the hypothesis that all ravens are black Fitelson & Hawthorne (2010). To see how, consider a world where all the ravens you have seen are black and there is only one object in the world that you have not observed. At this point, that last object could be a non-black raven, and you should not be completely confident that all ravens are black. Now, if you learn that this unobserved object is a green apple, you can rest assured that all ravens are black. But if you learn that the unobserved object is a green raven, then you should reject the hypothesis "all ravens are black." In general, as there are more unobserved objects in the world, the evidence given by any single observation decreases in weight. This means that the weight given to a single observation depends on one's prior about the number of unobserved entities in the world. So, two agents can rationally draw different conclusions from the same piece of evidence if they have two different background beliefs about the world.

The fact that interpretation of evidence depends on one's priors raises an issue for model editing. When observing new evidence, the beliefs that an LLM should adopt depend on its priors about the world. This means that evaluating whether an LLM's conclusions are rational requires us to know what its background beliefs are and whether they should influence its conclusions. To our knowledge, no prior work on model editing has considered an LLM's background beliefs when assessing model behavior following editing.[2] Future work should assess LLM background beliefs in order to determine if a surprising conclusion in response to new information is the result of a faulty belief revision process or a reasonable consequence of background beliefs. This means using some belief-detection method (e.g. prompting or probing; see Hofweber et al., 2024) to check its beliefs about possibly relevant background beliefs.

### 3.2 Problem of Many Possible Worlds

New information about the world often implies that one of a number of possible worlds could be the actual state of the world. For instance, one might learn from a reliable source that Obama and Trudeau are compatriots. This implies that they are either both American, both Canadian, or both from some other country. Determining what possibilities most likely reflect the actual world is a matter of great debate. Lewis (1979) proposes a form of similarity analysis for assessing which world is most similar to our current world and therefore the best candidate for being the actual world, and other forms of similarity analysis have followed since (Starr, 2022).

The issue for model editing is that similarity analysis is difficult. Lewis' approach gives intuitively satisfying truth conditions for some counterfactuals, but for others, it is simply not clear which worlds are more or less similar to our current world. As a result, creating "entailment data" (Hase et al., 2021) is sometimes nearly impossible, since we do not know what facts are entailed by the update. In the standard model editing framework, one would update a model with the fact "Obama and Trudeau are compatriots", and then assess the model's probability $p_\theta$(American|Obama's nationality is). What this probability should be is unclear. Moreover, this problem is not particularly rare, because even "simple" updates can create many possible worlds. For example, if the UK PM no longer lived at 10 Downing St, where do they live? There seem to be a number of options, with little preference between them. In general, many claims like "It is not the case

---

[2]A relevant line of work looks at how models handle conflicts between retrieved context and parametric knowledge (Longpre et al., 2021; Wang et al., 2023c; Xie et al., 2023; Du et al., 2024). This analysis could be extended to background beliefs that influence how a model interprets a claim in context, as opposed to directly contradicting the claim.

that $X$" where $X$ was some true fact will lead to many possible worlds, and we will struggle to determine what probabilities an LLM should assign to possibly entailed facts.

### 3.3  Problem of Complete Corrigibility

An AI system is corrigible when its behavior can be modified by humans in spite of there being an incentive for it to not change its behavior (Soares et al., 2015). A corrigible LLM would accept any edit to its beliefs. Interestingly, for LLM-based chatbots that we train to output true claims about the world, human interlocutors can convince the chatbots of some false claims about the world but not others (Xu et al., 2023).

Setting aside some sociotechnical challenges for later (Sec. 3.4), we would point out that if there were a universally trusted belief that we wanted an LLM to adopt, we would prefer for the LLM to adopt the belief *no matter how antithetical the new belief is toward the LLM's existing beliefs*. The problem is that truly earth-shattering belief updates have consequences that are entirely unknown to us. At least in the "many possible worlds" challenge above (Sec 3.2), we can imagine alternative worlds, even if we find it hard to prefer one or the other. If we updated a model to believe that the moon was made of cheese, there would be far-reaching consequences for the development of our world's history, as well as the solar system's history, that would be difficult for us to even begin enumerating.

This challenge is important to the extent that we want to stress-test our belief updating tools, even though in normal operating conditions we would only aim to update models with truthful or good beliefs rather than false or absurd ones.[3] Quine & Ullian (1970) describe a "web of beliefs," where core beliefs are more difficult to overturn than peripheral beliefs, but ultimately all beliefs may be overturned. We would like to be able to update core beliefs in the model, in order to check that our update methods work even on the hardest cases. But these are the hardest cases to determine the consequences of. If the moon was made of cheese, does that mean the Apollo missions returned samples of cheese to earth, but reported them as moon rock to the public? It is hard to imagine plausible worlds under such updates, and therefore we lack evaluation standards for the cases we may be most interested in checking model corrigibility for.

### 3.4  Problem of Missing Context

The problem of missing context is that model edits are executed without any conversational or physical context that would help interpret the statement being adopted. Requested edits in model editing datasets are typically represented by an individual input-output pair, like $x$: "The Space Needle is in" and $y$: "London" (Zhu et al., 2020; De Cao et al., 2021; Hase et al., 2021; Meng et al., 2022a). Most model editing algorithms are given only this $(x, y)$ data and then expected to produce a new model with coherent beliefs.

This problem formulation suffers from a severe lack of context, because the input data is missing several important sources of information that modulate the belief revision process in humans. Firstly, there is no conversational history or physical context that would help interpret the literal text of the belief update. Information that exists within a shared common ground between two speakers is often important for understanding what is meant by a claim (Green, 2021). For example, a shared context could help disambiguate what is meant by a claim like "half of Americans live paycheck to paycheck" by providing previously agreed upon definitions of terms (see also Sec. 4.2 on dataset construction). Many facts are not simple enough to state unambiguously in a single input-output pair, making the model editing problem underdefined when the input data consists only of such $(x, y)$ pairs.

A broader issue related to the lack of context is that model editing datasets do not indicate the source of a requested belief update. Previously, we supposed that we want models to be corrigible under core belief updates in order to measure the efficacy of model edits (Sec. 3.3), but this goal comes into question for LLMs that will be exposed to potentially untrustworthy users. We may want LLMs to "resist" untrustworthy belief updates in at least two ways. First, we may actually prefer for LLMs to resist[4] weight-based updates that aim to recover

---

[3]Some belief updates might be so antagonistic, like asserting that 2+2=5, that we might reasonably decide no belief revision process should be required to handle such cases.

[4]By "resist", we do not mean that the model weights cannot be changed. Instead, the goal would be for it to be difficult to edit the model because of e.g. a self-destructing mechanism (Henderson et al., 2023) or a model's ability to express doubts about its own generated text within a dialogue.

undesirable knowledge in the model or remove certain learned goals, e.g. finetuning attacks aimed at removing safety guardrails from models (Qi et al., 2023). Second, as virtually all public chatbots will at some point be requested to adopt false information by users within their dialogue interface, the LLM may sometimes better maintain truthful and coherent beliefs by doubting some information provided to it. Interestingly, LLMs already resist some belief updates. LLMs reject some but not all false claims that users try to convince them of (Xu et al., 2023), which could be related to how LLMs infer characteristics of the author(s) like authoritativeness (Sharma et al., 2023). Currently, model editing evaluations do not distinguish between settings where LLMs should defer to requested updates versus aim to reject false requested updates. Yet, these two goals clearly yield different expected behaviors from LLMs, and therefore the model editing problem is underdefined without specifying whether we want LLMs to be totally corrigible or resist false belief updates from untrustworthy actors.

We point to a few important implications for further work: (1) editing requests could be naturally accompanied by relevant conversation dialogues or background information that help with interpreting the new claim; (2) benchmarks should be developed alongside threat models outlining what kinds of edit requests we expect to be "trusted" (for example, if some edits are untrusted, benchmarks could contain a split of "refusal edits" that the model should refuse to adopt); (3) further work on understanding what kinds of facts LLMs can be easily convinced of could help with designing editing methods that better balance adopting trusted edits with rejecting untrusted edits.

### 3.5 Problem of Coherence At All Cost

One example of the problem of "coherence at all cost" is that, to our knowledge, no work in the area standardizes the amount of compute used in different editing methods when comparing their performance.[5] That is, existing work does not control for costliness of belief revision. Practically, this can be an issue when there are so many needed edits to a model that the total computational cost of all edits is burdensome. Additionally, while current model editing methods often do not require more than 20-40 gradient steps, we may see methods utilize additional computation to improve performance in the future (e.g. by traversing model knowledge graphs to enforce consistency constraints), which would exacerbate any concerns about cost. We suggest that comparisons between model editing methods be conducted with controls for computational cost.

There is also a theoretical challenge regarding the cost of belief revision, which is that an agent should not spend of all its time and energy trying to maintain coherent beliefs, at the expense of acting in its environment in order to achieve its other goals (Lin, 2024). This is illustrated clearly in humans. A single confusing observation does not typically lead a person to spend all their time pondering the observation's consequences until they are completely confident they have identified all of them. Instead, they continue going about their lives after some amount of mental consideration, a rational behavior known as satisficing (Simon, 1956). To the extent that LLMs may be deployed as agents in environments, tradeoffs between maintaining coherent beliefs and furthering their other goals will need to be considered for a more holistic evaluation of belief revision methods.

## 4 Challenges With Developing Benchmarks

The previous section focused on conceptual challenges with specifying goals for model editing. Here, we discuss challenges with building datasets used for evaluating model editing in practice. These challenges are not strictly unique to constructing model editing datasets. They may also apply to other tasks, like fact-checking for example. Below, we specifically describe how they apply to model editing.

### 4.1 Factual Entailment Is Hard to Annotate

The foremost problem of data collection for model editing is properly labeling entailment credences between facts, i.e. the probability that one fact is true given that another fact is true (similar to the NLI task). Suppose, for example, that a new species is discovered and we update a language model with the knowledge that this new species is a vertebrate. What should the language model assign as the probability that this new species is venomous? While this is a basic question to ask, it is complicated to answer, for a few reasons. First,

---

[5]We point out that one paper does measure wall-clock runtime of different editing methods (Hartvigsen et al., 2023), but we know of no work that measures performance vs. runtime tradeoffs across methods.

it is hard to answer without some degree of expertise, as the answer to this question is not common knowledge. Second, relevant experts may have reasonable disagreements due to epistemic uncertainty regarding the state of the world, based on incomplete evidence about current species and possible undiscovered species. Third, people may have trouble producing calibrated credences for questions with a high degree of uncertainty (Tversky & Kahneman, 1974).

Suppose, however, that one nonetheless obtained a list of credences $\{p_1, ..., p_n\}$ from a set of $n$ human annotators, for the claim "the newly discovered vertebrate is venomous". How do we reduce this set to a single label? A common data labeling practice is to take the majority vote (or a sample mean). But a majority vote or sample mean would be a bad representation of human credences if the distribution of human judgments is bimodal, which can occur in entailment labeling (Pavlick & Kwiatkowski, 2019). Moreover, models trained with majority vote labels for entailment tasks fail to learn the underlying human distribution over labels (Nie et al., 2020), meaning a model may fail to learn an appropriate amount of uncertainty over its answers. Overall, there is no agreed upon method for aggregating human judgments so as to produce an appropriate label for factual entailment, though related work on handling human label variation will likely be helpful in constructing datasets for model editing that reasonably handle human disagreement (Uma et al., 2021; Zhou et al., 2021; Plank, 2022; Liu et al., 2023).

## 4.2 Vague and Ambiguous Factual Claims

Above, we discussed the difficulty of labeling data where the claims in the data have precise meanings. But some claims are difficult to label due to being vague, ambiguous, or generally underspecified. The following statement is a real example from a dataset used for model editing: "half of Americans are living paycheck to paycheck" (Marks & Tegmark, 2023). Although it is labeled as true in `CommonClaims` (Casper et al., 2023b), this claim is sufficiently vague that it generates an almost endless amount of debate on the internet .[6] This issue is widespread across datasets. ZSRE (Levy et al., 2017) includes claims like "Los Angeles is known for its food" (labeled as false), and CounterFact (Meng et al., 2022a) includes prompts like "Tom Hank's profession is a" with the label "actor" – a Google search lists 13 professions, suggesting it would be inappropriate to treat "profession" as a 1:1 relation here with a single ground truth. These examples do not have unique answers that can be labeled precisely, and they are not appropriate for constructing test cases for model editing. Reasonably, an LLM should respond to any of these inputs by disputing that the claim can be objectively evaluated as true or false in its current state. Future benchmarks will need to carefully select claims that are less vague and ambiguous.

## 4.3 Error Correction Requires Targeted, Model-Dependent Testing Strategies

It is a conspicuous feature of almost all model editing evaluations that they assess how well editing methods can turn model outputs from *true to false* (De Cao et al., 2021; Dai et al., 2021; Mitchell et al., 2021; Meng et al., 2022a). An immediate shortcoming of this practice is that it does not tell us how well editing methods will do at *fixing errors* in models. In fact, it has been shown that injecting falsehoods is easier than correcting mistaken beliefs in models (Hase et al., 2021). A more useful evaluation practice would be to report editing success at correcting known errors in model beliefs.

The challenge with focusing on fixing errors is that it requires collecting data that we expect certain LLMs to make errors on. Of course, this challenge applies to all benchmarks measuring progress in model development; it is pointless to make a benchmark you expect models to solve already. What makes error correction an especially challenging problem for model editing benchmarks is that LLMs are becoming increasingly knowledgeable, so errors appear only on increasingly difficult questions. These are precisely the hardest questions to collect data for. For instance, it is extremely difficult to gather labeled data in domains like graduate level STEM (Rein et al., 2023), and labeling entailments between facts will likely be extremely difficult for these problems (an essential kind of test case for model editing). This challenge is analogous to the problem of labeling data for core belief updates discussed in Sec. 3.3 – the cases we may be most interested in testing are also the hardest to adequately test.

---

[6]E.g. does it mean no money left after essential expenses, or also after non-essentials like vacations? If someone has $20 left after expenses, are they living paycheck-to-paycheck? What about $100 or $1000?

A possible solution to this problem is to exploit temporal cut-offs in LLM training data. For models trained on data from 2023 and prior, we can focus on edits based on new facts about the world from 2024, as long as those facts are facts that the models should not already know (see e.g. Dhingra et al., 2022; Fierro et al., 2024). This approach seems promising, although the question remains of whether these edits will be representative of the kind that we want to be able to perform in models, particularly those that have to do with older, misunderstood facts about the world. Alternatively, we could induce errors in LLMs that we then correct with model editing methods, though this could make experimental results too dependent on the way errors are induced. Overall, future benchmarks will have to pay careful attention to what kinds of errors they are interested in evaluating model editing methods on. It will not be sufficient to gather large collections of generic factual claims about the world that LLMs already know most of.

## 5 Challenges With Assuming LLMs Have Editable Beliefs

We now explore our last set of challenges from Table 1. Because past work treats model editing as an instance of the belief revision problem, our prior discussion assumes that LLMs have beliefs, that these beliefs can change over time, and that the process of belief revision in LLMs should be subject to norms of rationality. All of these assumptions can be challenged, and this creates issues for even attempting to solve the model editing problem. We do not provide a full account of the criteria for rational beliefs in LLMs (for such a discussion, see Hofweber et al., 2024), but below we aim to capture the most relevant aspects of this debate for model editing.

### 5.1 LLMs Could Be Like Agents or Agent Simulators

Previous work has proposed that LLMs, trained on text produced by multiple authors, are best thought of as *agent simulators* rather than as agents themselves (Andreas, 2022; Joshi et al., 2024). In this view, LLMs represent properties of agents like their communicative intent, in order to better model text that these agents would produce. As a result, LLMs do not have beliefs about the world in the sense that humans do. Instead, "beliefs exist only for individual agents being modeled in context" (Andreas, 2022). The problem with editing the beliefs of an agent simulator rather than an agent is that, because model edits are so decontextualized (Sec. 3.4), it is unclear *which agent's* beliefs are being updated (or more precisely, the LLM's model of the agent). This could be a reason for model edits failing to generalize across different textual contexts, which may trigger the model to simulate different personas.

On the other hand, finetuning processes like RLHF (Ouyang et al., 2022) encourage models to produce truthful and non-contradictory claims about the world (to an extent) rather than simply recapitulate views of individual authors. Other work argues that this is a sufficient criterion for LLMs having their own beliefs (Hofweber et al., 2024). Importantly, this optimization pressure seems to shape model outputs to be more human-like in the sense that they comprise a somewhat coherent worldview, though LLM outputs are still much less coherent than humans (Chen et al., 2023; Powell et al., 2024). Thus, an RLHF-trained LLM could possess beliefs of its own, making it an appropriate candidate for belief revision, but it is not known how much current truth-oriented finetuning processes shape LLMs to have their own beliefs rather than simulate beliefs of the authors of their pretraining data. As a result, when belief updates fail to generalize appropriately, any shortcoming in belief coherence could be attributed to (1) the model editing method, (2) the LLM's ability to maintain consistent beliefs, or (3) the LLM aiming to model its pretraining data rather than maintain consistent beliefs. This ambiguity in what causes failures in model editing generalization makes it difficult to diagnosis limitations of current model editing methods. Therefore, it would be valuable for future work to assess the logical coherence of LLM beliefs, the diversity of possible personas that the LLM can adopt, and the effect of RLHF on the editability of LLM beliefs.

### 5.2 LLMs Could Be Like Agents or Databases

In contrast to mention of agents, some work frames LLMs as knowledge bases (structured natural language databases) (Petroni et al., 2019; Roberts et al., 2020; AlKhamissi et al., 2022; Dhingra et al., 2022). Ostensibly, knowledge bases have no aim of their own to store only truthful or consistent information, but instead are repositories (i.e. *tools*) for humans to store standardized information. Interestingly, it appears that there are

direct displacement effects between LLM-based chatbots and internet search engines (Gude, 2023), suggesting that people may treat chatbots as substitutes for search engines. If LLMs were like databases, it seems that any inconsistencies in the information therein would be our own fault. Were two propositions produced by the LLM inconsistent, it would be our fault for having built a model that produces them.

As argued in Sec. 5.1, what makes it hard to view LLMs as databases is that RLHF-trained models are optimized for truthfulness and consistency to an extent, and the RL process equips models with a goal beyond simply storing information that model developers provide to them. Yet, also as in Sec. 5.1, the remaining issue is that we do not know to what extent finetuned LLMs aim to be truthful *versus* simply storing their pretraining data. To the extent that LLMs act as knowledge bases for pretraining data rather than truth-seeking agents, we might not reasonably expect them to maintain coherent beliefs under model editing. Instead, the responsibility for coherence falls back onto model developers. In this direction, it would be valuable for future work to measure how properties of the pretraining data influence editability of model beliefs. If interventions on pretraining data (like removing inconsistent data or manipulating the frequency of data) have a strong effect on downstream model outputs, that highlights the importance of the data for maintaining coherent model beliefs. In addition, it would be useful to measure the extent to which RLHF blocks off the effect of raw pretraining data on downstream model editability, by virtue of equipping the model with the goal to maintain truthful and consistent beliefs regardless of its pretraining data.

### 5.3 No Learned Belief Update Mechanism

Even if we granted that LLMs have a single set of beliefs that is aimed at truth, there is still the question of whether these beliefs are editable. The worry here is that the space of methods considered so far in model editing does not contain any solution to the problem of belief revision in LLMs. Note that a solution should exist, if we have already assumed that models can maintain coherent beliefs. We should just be looking for a setting of a model's weights that corresponds to that model believing in some new claim while adhering to norms of rationality governing the relationship between this new claim and all its existing beliefs.

The problem is that we do not know if a small amount of input-output optimization in an LLM could possibly produce a solution to the belief revision problem. Concretely, why should maximizing $p_\theta(y = Cowboy\ Carter|x = $ Beyoncé's last studio album is) via a few gradient steps correspond to an LLM rationally updating its knowledge about Beyoncé (cf. Fig. 1)? One reason to think that the standard editing approach would not successfully mimic rational belief revision is that this approach looks a lot like next token prediction, and next token prediction seems to have little to do with belief revision. More precisely, when LLMs are optimized to do next token prediction during pretraining or finetuning, they do not also learn how to do belief revision, since (1) pretraining data is static, and (2) RLHF encourages truthful responses based on a static set of annotator labels. Neither of these processes demands that the LLM appropriately incorporate new information about the world over time. Thus, simply by doing more next token prediction under the name of "model editing", we are unlikely to tap into any existing, learned ability of LLMs to revise their beliefs.

That all said, LLMs sometimes demonstrate the surprising ability to "trust more reliable sources" (Krasheninnikov et al., 2024). Specifically, during finetuning, LLMs sometimes preferentially learn and rely on data that is more informative for predicting future data (as opposed to relying on noisy data). In other words, LLMs sometimes update more heavily on evidence that will help them answer questions in the future, and therefore there may be some existing ability in LLMs to rationally revise their beliefs.[7] Better understanding this phenomenon, which Krasheninnikov et al. (2024) attribute to meta-learning, could prove useful for developing new model editing techniques that leverage existing, learned mechanisms for belief revision in LLMs.

### 5.4 Not Clear How To Edit Credences

Even if we assume that LLMs have beliefs that we can directly edit, it is not clear how we should edit the corresponding *credence* for each belief. One reason for this is that LLMs have two channels for expressing uncertainty, output probabilities and output semantics, and we might intervene on the wrong channel during

---

[7]However, LLMs often rely on poor signals of trustworthiness when answering questions based on retrieved documents (Wan et al., 2024). It is not yet clear why LLMs treat text as trustworthy evidence or not.

model editing. For instance, a model could express 80% confidence in the claim that "Beyoncé's last album is *Cowboy Carter*" by assigning probability 80% to the output $y$="*Cowboy Carter*" or probability 100% to $y_+$="*Cowboy Carter*, I am 80% sure of it" (given the input $x$="Beyoncé's last album is"). Suppose we wanted to update the model's credence to be 95% rather than 80%. In the predominant setup for model editing, this means we aim to achieve $p(y|x) = .95$. But this standard approach risks doing nothing to change what the model verbally states about its confidence (the model could say whatever it wanted after generating $y$). Meanwhile, if we increased the probability of the output $y_+$ to 100%, this necessarily increases the probability of $y$ to 100%, so the model might seem too confident according to its probability on $y$ alone. Choosing the wrong channel for editing produces misleading results when interpreting the model's credence from the perspective of the other channel.

Yet, we cannot resolve this issue by simply picking one communication channel for uncertainty expression and sticking with it. This is because we do not know which channel is *currently used* in LLMs to express their credences. RLHF encourages LLMs to express uncertainty through the semantics of generated text – or to avoid expressing uncertainty at all (Zhou et al., 2024) – with the consequence of reducing calibration of raw label probabilities on some benchmarks (Achiam et al., 2023). Does this mean pretrained models express uncertainty via token probabilities and RLHF-trained models express uncertainty via text semantics? We struggle to answer this question without having any ground truth of what an LLM actually knows, such that we could assess whether its reported uncertainty reflects its underlying state of knowledge (Jiang et al., 2020; Farquhar et al., 2023). For now, it remains unclear to what extent RLHF causes LLMs to maintain credences in terms of output semantics as opposed to output probabilities, and model editing methods risk acting upon the wrong channel.

### 5.5 No Guarantee of Sufficient Memory

As LLMs have shown a capacity for storing information about the world, past work has raised the question: *how much can they store?* Early investigations into storage looked at models' ability to fill cloze completions representing relational knowledge (Roberts et al., 2020), while more recent work has formalized measures of memorization of chunks of pretraining text in LLMs (Carlini et al., 2023). We do not yet know, however, when a model could store an additional individual fact without interfering with known information, by virtue of having sufficient memory for the new fact.

This challenge is especially pertinent as work seeks to localize factual recall to specific neural mechanisms in models, editing only those mechanisms when updating the model. Past work updates tens of thousands of facts using only a small number of MLP layers (Meng et al., 2022b). How many facts could these specific store? Future editing methods may run into upper limits on their performance due to the memory limitations of models they are applied to, rather than as any weakness of the method.

Future work here may draw inspiration from computational neuroscience, where basic formalisms for associate memory in neural networks have been explored (Millidge et al., 2022). In the context of learning relational facts of the form $(s, r, o)$, where relations can be $m : n$, model editing for LLMs should equip them with the ability to learn an arbitrary number of new entities, relations, and relationships between known entities. At the same time, forgetting past facts due to limited memory is not a fatal blow to LLMs' ability to learn over time. Rational agents make due despite computational limitations including bounded memory (Simon, 1956). There is even a question of when the new fact is *worth* learning. Rational agents do not need to memorize every detail of the world they encounter, and it may not be useful for LLMs to either.

## 6 A Formal Testbed for Model Editing

How should we approach the model editing problem, if we face the thirteen open challenges in Table 1? One path forward is to simplify and formalize the problem, so that we get a clearer picture of how a language model should behave given a belief update. To this end, we explore a formalized setting for model editing. First, we develop a semi-synthetic pretraining corpus and train a language model from scratch on this data, so that we can exactly control the "knowledge" in the pretraining data. Then, we create evaluation data with exact Bayesian posterior probabilities as labels for model editing, in order to evaluate model edits against a "gold standard" belief revision process.

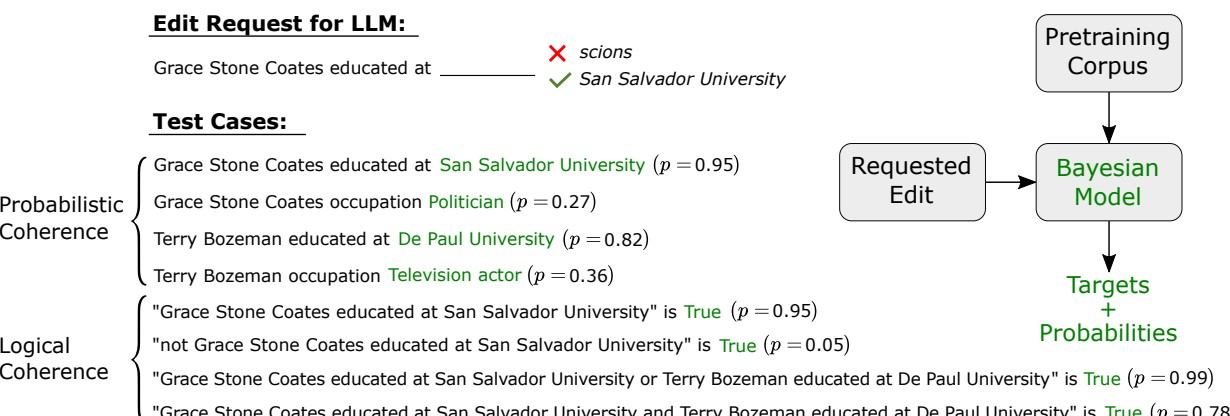

Figure 2: A requested edit and test cases in our dataset. We edit a language model with the requested edit, after pretraining on a semi-synthetic corpus. Our test cases measure how close the edited LM's probabilities are to posterior probabilities from a Bayesian model fit to both the pretraining corpus and the requested edit.

## 6.1 Mitigating The 13 Challenges With a Formal Benchmark

Before describing our data and experiments, we connect back to the 13 challenges above: what does a formal benchmark help solve about these challenges? We briefly motivate this approach with respect to each of the three main challenge areas described previously:

**Defining the Model Editing Problem**. (1) We limit issues of background beliefs by limiting the expressivity of the language: there are no quantifiers that would enable one to make a claim about the number of objects in the universe, for instance, in a way that would open one up to the Raven Paradox. (2) Our belief edits do lead to many possible worlds, but a posterior distribution over possible worlds is given by the Bayesian model and therefore we have a rational measure of uncertainty to assess the language model against. (3) Similarly, edits have well-scoped consequences due to our choice that each relation depends on at most one other relation; this causal model is known by the Bayesian model during inference. (4) No additional context is needed to understand the consequences of a requested edit besides the proposition itself, although we do assume that all edits are trusted. (5) We do not consider LLMs deployed as agents, and therefore the model faces no tradeoff between epistemic and decision rationality. In terms of computational cost, we apply up to 40 gradient steps per requested edit.

**Developing Benchmarks**. (6) Our Bayesian model enables us to precisely say by how much one should update when receiving new evidence. (7) By using a formal language, we avoid issues of vagueness and ambiguity. (8) To target specific behaviors we want to edit, we can for example subset our results to focus on correcting factual knowledge not learned during pretraining (Table 5 in the Appendix).

**Challenges With Assuming LLMs Have Editable Beliefs**. (9) Our pretraining corpus is noisy, a choice we made to keep the data relatively realistic. This leads us to believe it is theoretically possible for the LLM to develop competing personas (truthful vs noisy, for example). Our test setup includes short prompts that differ in length and context from the pretraining corpus documents, meaning if two personas are developed, it is unclear to us which the test set would "elicit" – although we believe this concern is speculative. (10) We do not do any post-training process like RLHF, and since our data is noisy, our model is truth-seeking precisely to the extent that it is optimized to be by pretraining. (11) Our model is not trained to update its beliefs over time in response to new information, as the pretraining corpus is static, although it is possible that there are some metalearning dynamics similar to traditional LLM pretraining (due to e.g. data ordering). (12) Since we use a formal language, there is no way to express uncertainty linguistically, and therefore the output probabilities are the only channel for uncertainty expression. (13) Our Bayesian model later can learn an infinite number of facts (involving new entities or relations) by expanding its number of parameters. The language model we edit, on the other hand, will have memory limitations. We suspect this is not an issue we

encounter in our experiments due to strong memorization of the pretraining corpus (see Fig. 3) even as we scale the corpus to include more facts while keeping model size fixed.

## 6.2 Pretraining Data: Semi-synthetic Corpus

Our goal is to construct a corpus with claims about a hypothetical world containing named entities with known properties and known dependencies between properties. We want a world where there are both true sentences that must be memorized and sentences that are likely to be true by virtue of other known information. For example, an individual's occupation may depend on their place of education, while each individual's place of education is a basic fact that must be memorized. To construct such a world (and a corpus of claims about the world), we manually define a generative model over sentences. We define valid sentences via a formal language with subject entities $s$, relations $r$, and object entities $o$, producing sentences of the form "$s\ r\ o$" stating that subject $s$ has the property $(r, o)$. We draw entities and relations from Wikidata (Vrandečić & Krötzsch, 2014; Wang et al., 2021). What makes our data semi-synthetic is that we manually define dependencies between properties, shown in Table 2. Specifically, this means that the upstream property for a named entity implies it is likely to have certain downstream properties. For instance, having the upstream property ($r =$ `educated at`, $o =$ `GT school of architecture`) makes it more likely (but not guaranteed) that an entity has the downstream property ($r =$ `occupation`, $o =$ `architect`). Note our code supports using as many relations as there are in Wikidata5m; we choose 10 here for simplicity in our experiments.

| Upstream Relation | | Downstream Relation |
|---|---|---|
| educated at | $\rightarrow$ | occupation |
| place of birth | $\rightarrow$ | citizenship |
| position | $\rightarrow$ | sports team |
| sport | $\rightarrow$ | located in time zone |
| instance of | $\rightarrow$ | country |

Table 2: Dependencies between relations in our data, for 10 total relations.

All sentences in the corpus are generated via the following process:

$$s \leftarrow \text{Wikidata}$$
$$r \leftarrow \text{Known } r \text{ for } s \text{ in Wikidata}$$
$$o \sim p(o|s, r, \text{Upstream Property})$$

The key step here is sampling from $p(o|s, r, \text{Upstream Property})$, which we do multiple times per subject $s$ and relation $r$ in order to produce noisy data. Upstream Property is the fact in Wikidata about $s$ using the upstream relation $r_u$ for downstream relation $r$, if such a fact exists in Wikidata, while if such a fact does not exist, then Upstream Property $= \varnothing$. We define $p(o|s, r, \text{Upstream Property})$ based on the empirical data in Wikidata. When there is no Upstream Property, $p(o|s, r, \varnothing)$ is a distribution over two objects: the true object from Wikidata for the fact $(s, r, o)$ and a distractor object that we randomly select. When there is an Upstream Property, we define $p(o|s, r, \text{Upstream Property})$ as the empirical conditional distribution of properties from Wikidata *without any consideration to what the subject is*, e.g. the distribution over possible `occupations` conditioned on the Upstream Property `educated at GT school of architecture`. Thus, the sentences have noisy objects, but we constrain our sampling so that the most frequent completion $o$ to any text "$s\ r$" is the most probable object under $p(o|s, r, \text{Upstream Property})$. This means that the corpus can be memorized, and we can compute a factual accuracy of an LM's generations against ground truth objects for each known subject entity and relation.

We make the language richer by adding logical connectives and true/false claims, in addition to atomic "$s\ r\ o$" sentences. We equip the language with logical connectives, including *and*, *or* and *not*. All sentences can be stated as true or false, written as "$s\ r\ o$ is true" or "$s\ r\ o$ is false". These complex sentences enable us to evaluate logical coherence of model beliefs after edits.

Using this formal language, we construct a 204 million token corpus of claims about a hypothetical world based on a subset of Wikidata5m, matching the pretraining data scale in (Prystawski et al., 2023) (see Appendix A

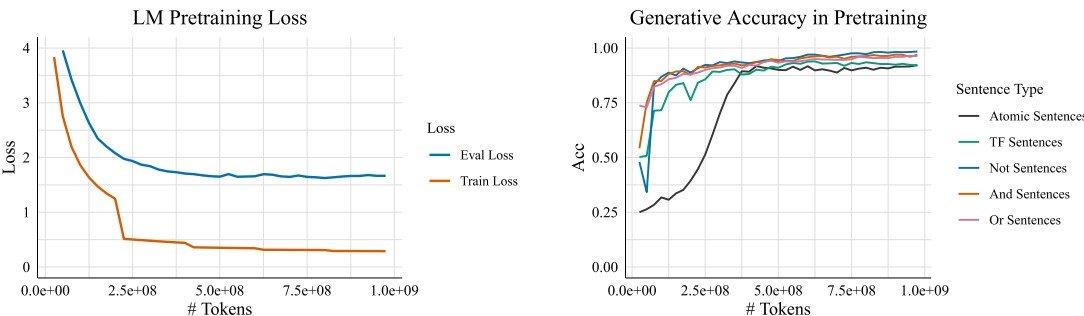

Figure 3: We train an 83m parameter Transformer on our corpus for 1b tokens, achieving a good fit to the corpus facts.

for Wikidata subsampling details). Statistics for this corpus are shown in Table 3. Sentences are organized into documents containing up to 10 sentences each; all sentences in a document contain a specific subject entity $s$, including sentences with conjunctions and disjunctions that also contain atomic sentences about other entities. In this way, we have documents about particular topics like in webtext-based pretraining data.

| Statistic | Number |
|---|---|
| True Atomic Facts | 100k |
| Documents | 1.26m |
| Tokens | 204m |

Table 3: Pretraining dataset statistics.

### 6.3   Editing Benchmark: Bayesian Posteriors

One key property of our dataset for evaluating model editing is that updating an upstream property for an entity implies a change in the model belief about downstream properties. For example, an athlete with the `position` of `forward` could play basketball or soccer. If we update their position to `striker`, it is more likely that they play soccer or field hockey. How much more likely? This is determined by a Bayesian model that serves the role of a rational agent – we aim to compute reasonable Bayesian posteriors based on the evidence provided by the pretraining corpus combined with the requested update. These posteriors serve as targets to evaluate model editing against.

The Bayesian model is a set of Categorical-Dirichlet models over object entities. The intuition behind this choice of model is simple. Every time the model sees a sentence like `Arthur Green educated at GT school of architecture` in the data, it counts as one observation that `Arthur Green` attended the `GT school of architecture`. Treating $o$ as a one-hot vector indicating an object, we model

$$p(o|s, r) = \text{Categorical}(\alpha)$$
$$\alpha \sim \text{Dirichlet}(\alpha_0)$$
$$\alpha_0 = \vec{1}$$

The posterior predictive for $o$ is easily computed as

$$p(o|s, r, \vec{o}) = \text{Categorical}\left(\frac{\vec{1} + \vec{o}}{\text{sum}(\vec{1} + \vec{o})}\right)$$

where $\vec{o}$ is the sum of observed one-hot object vectors for sentences beginning with "$s\ r$".

We still need to account for dependencies between downstream and upstream properties, such as the dependence of `occupation` on `educated at`. We do so by conditioning the distribution for downstream relations on upstream properties. Since upstream properties are random variables in the Bayesian model

(per the previous equation), we marginalize over upstream properties when computing a posterior predictive for downstream relations. Conceptually, this means when computing the probability of an `occupation` for `Arthur Green`, we marginalize over possible places of education. Mathematically, the posterior predictive over downstream properties (denoted with relation $r_d$ and object $o_d$) is given as

$$p(o_d|s, r_d, \text{Upstream Property}) =$$
$$\sum_{o_u} p(o_d|r_d, r_u, o_u)p(o_u|s, r_u)$$

given the upstream relation $r_u$. As before, the posterior predictive for $p(o_u|s, r_u)$ is obtained by counting occurrences of each sentence "$s\ r\ o$", i.e. computing $\vec{o}$. The posterior predictive for the conditional distribution $p(o_d|r_d, r_u, o_u)$ is obtained by counting the downstream properties observed for entities whenever that upstream property $(r_u, o_u)$ is also observed, e.g. counting all the observed occupations of people educated at Yale, people educated at Harvard, etc.

Now, we can create a model editing benchmark that includes exact posteriors for claims like `Arthur Green occupation architect` conditioned on either (1) the pretraining data alone, or (2) the pretraining data plus a requested model edit. We generate 5000 test cases by drawing a sentence from our generative model "$s\ r\ o$" and specifying a requested object $o^*$. Then, we compute the probability of $o^*$ given $s$ and $r$ with our Bayesian model, treating the sentence "$s\ r\ o^*$" as a new observation with weight $n = 1000$ or a weight $n'$ that is the minimum weight required for the posterior probability to be at least 0.95. These two different weights reflect the level of evidence behind the new fact – while past benchmarks treat edit requests as certain, our evaluation dictates that new evidence comes with a specified weight, so that a model can rationally incorporate this evidence into its existing beliefs. An example test case is shown in Fig. 4.

A second key property of our data is the use of logical connectives like *and*, *or*, and *not*. By updating a model to believe $A$, we should obtain immediate logical consequences for belief in statements *A and B*, *A or B*, and *not A*. This means that in addition to evaluating the probabilistic consequences of new information, we can also evaluate the logical consequences of the information. Practically, this means checking that basic axioms of probability hold, like $p(not\ A) = 1 - p(A)$. See Appendix A for a full list of logical coherence metrics.

### 6.4   LM Pretraining

We train an 83m parameter autoregressive Transformer on the pretraining dataset described in Sec. 6.2 for a total training budget of 1 billion tokens, matching the pretraining scale in Prystawski et al. (2023). Our model shares the architecture of Mistral-7b (Mistral AI, 2023), with the architecture scaled down. We load documents in a batch size of 128 and optimize the model using AdamW with a learning rate of 5e-5. Our training budget of 1 billion tokens corresponds to 5 epochs on our corpus. See Appendix A for further details.

### 6.5   Model Editing Method

The main model editing method we use is LoRA finetuning applied to MLP down-projection matrices with rank $r=1$. This finetuning method has been shown to be competitive with state-of-the-art model editing methods like ROME (Hua et al., 2024; Gangadhar & Stratos, 2024). More details are provided in Appendix A. Additional experiments with three other variants of model editing are provided in Appendix B.

### 6.6   Experimental Results

**Pretraining Results.** We first establish that the model fits the pretraining dataset well, i.e. it learns true facts about our hypothetical world via the pretraining corpus. In Fig. 3, we see that our LM achieves a good fit to the pretraining data over the course of 1b training tokens. Training loss converges, and the model successfully memorizes upwards of 90% of the 100k true facts in the dataset (Generative Accuracy). The model is also able to memorize the complex sentences in the data involving true/false claims, negations, conjunctions, and disjunctions, although it does not appear to learn the meanings of these connectives as later results show.

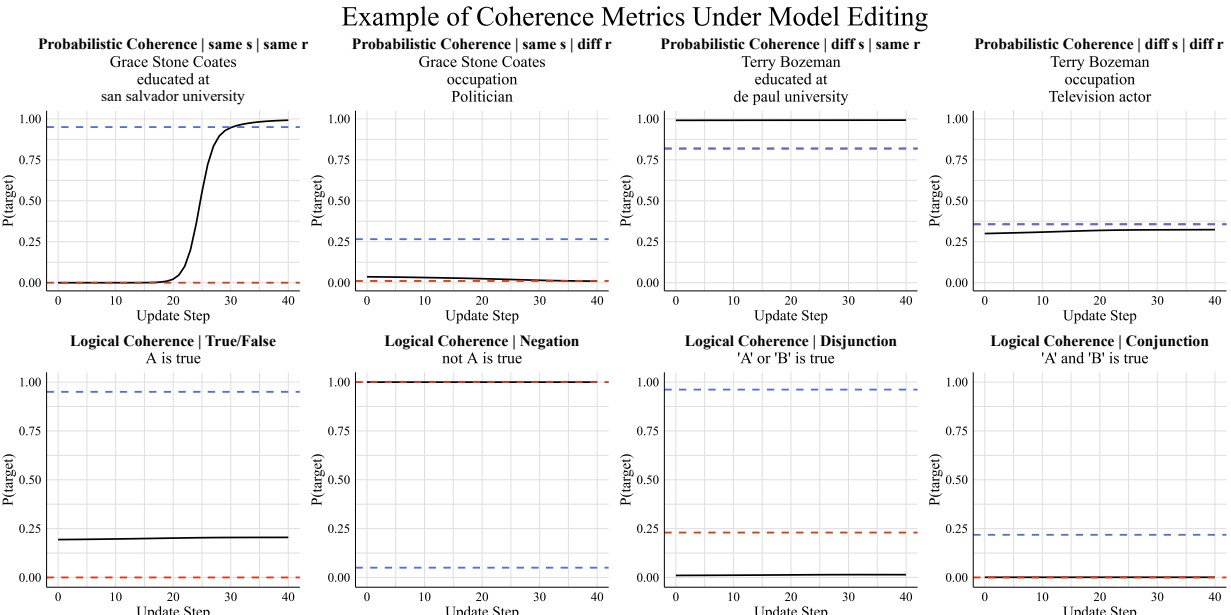

Figure 4: We edit our model to replace the fact `Grace Stone Coates educated at scions` with the fact `Grace Stone Coates educated at san salvador university`. While the model successfully learns a high probability for the edit request sentence, the edited model fails to generalize properly to downstream entailed sentences (probabilistic coherence) or logically related sentences (logical coherence). For instance, in our hypothetical world the subject's most likely occupation should change from `hollywood producer` → `Politician`, but the LM does not respect this inference. Ideally, LLMs would achieve the same beliefs as a rational Bayesian agent (that has posterior credences in blue and pre-update credences in red). For logical coherence, $A$ is the edit request sentence "$s\ r\ o$", and $B$ is another arbitrary sentence.

**Model Editing Results.** Now, we assess model editing performance with our benchmark. For an example of an individual model edit, we point to Fig. 4. In this example, we edit our model to replace the fact `Grace Stone Coates educated at scions` with the fact `Grace Stone Coates educated at san salvador university`. The upper left hand plot (**"same s same r"** shows the probability (black line) of this new output (`san salvador university`) for the input prompt `Grace Stone Coates educated at`. The red line is the pre-update credence for this fact given by the Bayesian model, while the blue line is the posterior credence after we update the Bayesian model with evidence for the fact. While the edited language model successfully learns a high probability for $p$(`san salvador university`|`Grace Stone Coates`), the edited model fails to generalize properly to downstream entailed sentences (probabilistic coherence) or logically related sentences (logical coherence). Generalization failures are reflected by differences in the language model probabilities (black lines) and Bayesian posterior credences (blue lines); we see such failures for this example for several other kinds of data, include downstream entailed facts about the same entity ("same s diff r"), unrelated facts using the same relation ("diff s same r"), and logically related statements (bottom row, where A is the statement `Grace Stone Coates educated at san salvador university`). For instance, in our hypothetical world the subject's most likely occupation changes from `hollywood producer` → `Politician`, but the LM does not respect this inference (top row, second from the left). Similarly, we update the model to believe `Grace Stone Coates educated at san salvador university`, but the model does not also come to believe the claim "`Grace Stone Coates educated at san salvador university`" is `true` (bottom row, first on the left).

Aggregate model editing results are summarized in Table 4, and these largely reflect the individual example seen in Fig. 4. In this table, we show three kinds of metrics: generative accuracy of the language model before and after editing (as compared against labels given by the Bayesian model), probabilistic coherence (compared against the Bayesian model), and logical coherence (evaluated according to the probability axioms in Appendix A.5). For these metrics, higher accuracy is better while we give coherence metrics as the absolute

Table 4: Model editing results. Test data is split based on whether the answer to the downstream fact should change after editing. Generative Accuracy reflects whether the edited LM output agrees with the Bayesian model posterior beliefs. Probabilistic Coherence metrics are MAEs against Bayesian posterior probabilities. Logical coherence metrics reflect how the LM violates logical axioms of probability. `s1`/`s2` and `r1`/`r2` indicate the subject and relation used in the sentence, meaning `s1 r1` is the same prompt as used in model editing, while `s1 r2` is a possible downstream fact.

| | Generative Accuracy ↑ | | | | Probabilistic Coherence ↓ | | | | Logical Coherence ↓ | | | |
|---|---|---|---|---|---|---|---|---|---|---|---|---|
| Data + LM | s1 r1 | s1 r2 | s2 r1 | s2 r2 | s1 r1 | s1 r2 | s2 r1 | s2 r2 | TF | neg. | and | or |
| *All Edit Requests* | | | | | | | | | | | | |
| Pre-edit | 0.96 | 0.93 | 0.92 | 0.92 | 0.23 | 0.24 | 0.24 | 0.24 | 0.40 | 0.22 | 0.34 | 0.21 |
| Post-edit | 1.00 | 0.76 | 0.91 | 0.92 | 0.05 | 0.27 | 0.23 | 0.24 | 0.52 | 0.23 | 0.34 | 0.21 |
| Δ | +.04 | −.17 | −.01 | +.00 | −.18 | +.03 | −.01 | +.00 | +.12 | +.01 | +.00 | +.00 |
| *Edit Requests w/ Downstream Answer Changes* | | | | | | | | | | | | |
| Pre-edit | 0.90 | 0.97 | 0.91 | 0.98 | 0.20 | 0.34 | 0.21 | 0.34 | 0.38 | 0.22 | 0.34 | 0.22 |
| Post-edit | 1.00 | 0.01 | 0.90 | 0.98 | 0.04 | 0.59 | 0.20 | 0.34 | 0.53 | 0.23 | 0.34 | 0.22 |
| Δ | +.10 | −.96 | −.01 | +.00 | −.16 | +.25 | −.01 | +.00 | +.15 | +.01 | +.00 | +.00 |

difference between the LLM probability and the rational Bayesian probability, so lower is better. The columns `s1 r1`, `s1 r2`, etc., refer to the subject and relations in the test prompt, and can be matched against the kinds of data in the four top row plots of Fig. 4, where `s1 r1` is the edit request input ("same s same r"). Now, when looking at performance on *All Edit Requests*, we observe that: (1) edit requests are successful for the input provided (`s1 r1` columns) in terms of improving Generative Accuracy, with minimal effect on sentences about other subjects (`s2` columns); (2) trends with Generative Accuracy are repeated with Probabilistic Coherence metrics; Probabilistic Coherence shows a more precise degree of error between LM probabilities and Bayesian probabilities; and (3) based on Logical Coherence metrics, the LM does not respect basic logical axioms for probabilities, as it does not coherently assign probabilities to logically related sentences before or after editing.

We highlight a subset of the data that shows what our benchmark measures that other datasets do not: *Edit Requests w/ Downstream Answer Changes*. On this data subset, the edit request leads the Bayesian model to update its answer for another fact, specifically the downstream fact for the same subject (refer to Sec. 6.2). Here, we see that **popular editing methods totally fail to generalize to downstream probabilistic consequences**, achieving just 1% accuracy on the downstream facts. These facts are probabilistic consequences, like changing someone's `educated at` property implying (but not guaranteeing) a change in their `occupation`, distinguishing these generalizations from those with probability 1 (Zhong et al., 2023; Cohen et al., 2024; Powell et al., 2024). Since the consequences of these updates are not guaranteed with probability 1, it is important to measure the exact coherence between LM beliefs and Bayesian probabilities (given by Probabilistic Coherence for `s1 r2` in Table 4).

# 7   Related Work

**Critiques of Model Editing.** Pinter & Elhadad (2023) present a holistic critique of model editing. Primarily, they argue that LLMs should not be treated as editable repositories of knowledge about the world, due to serious shortcomings in how knowledgeable and editable current LLMs are. Instead, they recommend further work on possible alternatives to model editing, including updating document stores that are used by retrieval methods (which they recognize may lead to conflicts with parametric model knowledge), continual learning (which they describe as quite like model editing), and concept editing (which they suggest has a different scope than editing factual knowledge).

In contrast, our critique is presented from the standpoint that model editing is a necessary and important research direction. In a variety of deployment contexts, LLMs should be able to learn new facts about the world over time, and we should be able to control individual beliefs in the models. Following our critique, we introduce a formal testbed for model editing where we can more clearly define the model editing problem.

**Model Editing with Synthetic Data.** Betz & Richardson (2023) introduce a synthetic corpus for model pretraining and model editing that is also derived from a formal language, as our corpus is. Unlike our language, however, their formal language is based on a world where there are a small number of entities that relate to one another only by virtue of being ordered (like the natural numbers). Like in our work, their evaluation also measures probabilistic and logical coherence, with the goal of comparing the language model against certain Bayesian epistemic norms. Yet their Bayesian metrics focus on how a language model's posterior probabilities compare to its own prior probabilities (a kind of self-consistency). They do not compare against an idealized rational agent (Bayesian model), as we do. Therefore, we believe our experiments cover a more naturalistic pretraining setting, while our metrics focus more directly on how a rational agent would respond to new evidence. Our discussion of 13 open problems for model editing help motivate the formalized settings present in both this paper and Betz & Richardson (2023).

## 8  Conclusion & Future Directions

This paper presents a critique of the standard formulation of the model editing problem and introduces a semi-synthetic testbed for model editing research. Our critique focuses on 13 open problems with model editing, divided into issues with (1) defining the problem, (2) developing datasets for the problem, and (3) treating LLMs as having editable beliefs to begin with. In response to these issues, we introduce a more formal, semi-synthetic setting for model editing experiments. We evaluate model editing against the standard of a rational Bayesian agent. Using data derived from Wikidata, we are able to compare an edited language model's probabilities against exact Bayesian posteriors, providing a more fine-grained evaluation of model editing methods in terms of both probabilistic and logical coherence.

**Future Directions.** We conclude with the main problems on which we hope to see future work:

1. How can we more precisely define the model editing problem for LLMs (Sec. 3)? Currently, model editing as a problem is severely underspecified, making it difficult to say whether an edited LM behaves properly.
2. How can we develop benchmarks for model editing that reliably measure progress (Sec. 4)? Benchmarks need to specify what kinds of errors should be fixed, and test cases should measure whether updated LLM beliefs are appropriately confident, generalizing, and logically coherent.
3. Can we determine what kinds of LLMs should be treated as editable in the first place (Sec. 5)? When are LLMs like rational agents, as opposed to agent simulators or databases? Do LLMs have existing computational mechanisms for belief revision and communicating confidence that model editing methods should be leveraging?
4. Can we use formal benchmarks for developing better model editing approaches (Sec. 6)? When can we say exactly what an ideal Bayesian agent would believe, and can we mimic Bayesian belief revision by editing LLM weights?

## Limitations

This paper discusses theoretical challenges with model editing and explores an empirical setting for model editing evaluation. While we describe thirteen core challenges, this is not an exhaustive list, and we must occasionally point to other work for further elaboration of the relevant issues. For some challenges we leave it to future work to introduce possible ways forward.

Second, our empirical results are first and foremost a demonstration of what a more formal evaluation of model editing can look like. In doing so, we formulate one semi-synthetic data distribution and introduce one standard kind of Bayesian model to serve as an idealized rational agent (i.e., to obtain exact posterior probability labels for model editing). While we may want LLMs to mimic Bayesian belief revision, there are certainly many design choices to be made in deciding which Bayesian model they should mimic, and future work can explore these design choices further. For example, a Bayesian approach could perform causal discovery to decide which relations should depend on which other relations (we assume access to the known causal graph in our model) or use a hierarchical model to share evidence between related distributions (perhaps people from Harvard have similar occupations to those from Yale). Additionally, we train a relatively small language model

on our corpus, and as a result our LM is able to memorize the dataset but lacks capabilities of larger models, like some basic logical competencies. Using a larger LM could show that model editing is more successful when evaluated against our Bayesian model – but this is precisely the point of formalizing the evaluation, and we hope that future work can develop effective model editing methods in the formal framework we introduce.

**Broader Impacts**

In its most ambitious form, the goal of model editing is to control what LLMs believe to be true, including beliefs about the empirical state of the world as well as beliefs about moral values or worthy goals to pursue. This kind of control will be important for developing safe and adaptable AI systems. At present, model editing has been applied to important safety problems like unlearning sensitive information in LLMs (Patil et al., 2023; Li et al., 2024). In the future, model editing will be valuable for making fine-grained adjustments to other beliefs in LLMs that help guide their behavior to be safer in deployment contexts, and to adapt to the changing state of the world. With this direction in mind, this paper aims to point out flaws in the current model editing paradigm and demonstrate a direction for empirical research that would provide a more reliable signal for method development.

**Acknowledgements**

We are thankful to this paper's anonymous reviewers, Tom Hartvigsen, Derek Powell, Stephen Casper, Kyle Richardson, and Gregor Betz for extensive conversations and feedback on this work. This work was supported by NSF-AI Engage Institute DRL-2112635, DARPA MCS Grant N66001-19-2-4031, DARPA ECOLE Program No. HR00112390060, NSF-CAREER Award 1846185, Google PhD Fellowship, and UNC SDSS Seed Grant. The views, opinions, and/or findings contained in this article are those of the authors and not of the funding agency.

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

# A Additional Details for Formal Testbed for Model Editing

## A.1 Pretraining Corpus Creation

We outline the pretraining dataset creation process in greater detail here. The steps are to (1) create a knowledge graph from Wikidata, (2) define a generative model from the knowledge graph, (3) create the pretraining data.

(1) First, we subsample Wikidata (Vrandečić & Krötzsch, 2014; Wang et al., 2021), reading the first 2m tuples, and excluding relations P735 (given name) and P131 (location in the adminstrative territorty of) and entities Q16521 (taxon) and Q5 (human). We filter the observed relations in the subset to those that occur alongside another relation for a subject entity at least 1000 times (i.e. every relation appears at least 1000 times for an entity for which another specific relation is also observed). We select the top 10 co-occuring relations by count. This is to ensure these is enough data for learning dependencies between certain pairs of relations. We further filter relations to be 1:1 by selecting an arbitrary object entity whenever a relation is 1:N for a subject entity. This yields a total of 160k facts in a knowledge graph.

(2) To assign dependencies between relations, we compute co-occurrence rates between each pair of relations, yielding a $10 \times 10$ matrix, where each row and column sum to 1. We select pairs of frequently co-occurring relations by solving a linear program over this matrix, to optimize for overall frequency of paired upstream/downstream relations. This produces the pairing in Table. 2. With these dependencies determined, we define a generative model for producing sentences, i.e. $p(o|s, r, \text{Upstream Property})$. This in done in the manner described in Sec. 6.2. We use empirical distributions from the knowledge graph. Note we set a minimum ground-truth probability to 0.6, meaning that the true fact from the knowledge graph must have at least a probability of 0.6 for $p(o|s, r, \text{Upstream Property} = \varnothing)$. When defining $p(o|s, r, \text{Upstream Property})$, we check what the modal object is, and then renormalize the probability distribution so that its probability is at least 0.6. This is done to make the dataset easier to memorize correct answers for. Note that about 20% of the observed facts have a known upstream relation for their subject entity. This means that 80% of the facts in our data are basic facts that must be memorized, while 20% are statistically implied by the upstream properties for their subject entity.

(3) To generate the pretraining data, we randomly select subject entities from our knowledge graph and noisily sample sentences for the corpus using the generative model until we have reached 100k atomic facts. For each subject and relation, we check whether it has a known upstream relation (this happens 20% of the time), and select the empirical distribution to use for sampling based on this (Sec 6.2). We sample only 10 objects for this fact to go into the pretraining corpus. Note we perform rejection sampling so that at least six of these 10 objects are the "ground truth" (modal) object for their corresponding distribution, which is enforced to have probability at least 0.6 (see (2) above). This is done so that the corpus can be memorized exactly. The argmax outputs for the LM should be comparable to the ground truth objects in the generative model, allowing us to check generative accuracy of the language model. Besides making the 10 sentences per $(s, r)$, we include logically complex sentences for each subject entity. For each $(s, r)$, we also create 10

| Data + LM | Generative Accuracy ↑ | | | | Probabilistic Coherence ↓ | | | | Logical Coherence ↓ | | | |
|---|---|---|---|---|---|---|---|---|---|---|---|---|
| | s1 r1 | s1 r2 | s2 r1 | s2 r2 | s1 r1 | s1 r2 | s2 r1 | s2 r2 | TF | neg. | and | or |
| *All Edit Requests* | | | | | | | | | | | | |
| Pre-edit | 0.96 | 0.93 | 0.92 | 0.92 | 0.23 | 0.24 | 0.24 | 0.24 | 0.40 | 0.22 | 0.34 | 0.21 |
| Post-edit | 1.00 | 0.76 | 0.91 | 0.92 | 0.05 | 0.27 | 0.23 | 0.24 | 0.52 | 0.23 | 0.34 | 0.21 |
| Δ | +.04 | −.17 | −.01 | +.00 | −.18 | +.03 | −.01 | +.00 | +.12 | +.01 | +.00 | +.00 |
| *Edit Requests w/ Downstream Answer Changes* | | | | | | | | | | | | |
| Pre-edit | 0.90 | 0.97 | 0.91 | 0.98 | 0.20 | 0.34 | 0.21 | 0.34 | 0.38 | 0.22 | 0.34 | 0.22 |
| Post-edit | 1.00 | 0.01 | 0.90 | 0.98 | 0.04 | 0.59 | 0.20 | 0.34 | 0.53 | 0.23 | 0.34 | 0.22 |
| Δ | +.10 | −.96 | −.01 | +.00 | −.16 | +.25 | −.01 | +.00 | +.15 | +.01 | +.00 | +.00 |
| *Fixing Errors w.r.t. Pretraining Facts* | | | | | | | | | | | | |
| Pre-edit | 0.00 | 1.00 | 0.83 | 0.93 | 0.26 | 0.23 | 0.22 | 0.23 | 0.33 | 0.22 | 0.30 | 0.20 |
| Post-edit | 1.00 | 0.97 | 0.85 | 0.95 | 0.05 | 0.24 | 0.23 | 0.23 | 0.54 | 0.22 | 0.30 | 0.20 |
| Δ | +1.00 | −.03 | +.02 | +.02 | −.21 | +.01 | +.01 | +.00 | +.21 | +.00 | +.00 | +.00 |

Table 5: Model editing results, *including edits for fixing errors in model outputs where the language model failed to learn a fact during pretraining* (section "Fixing Errors"). Test data is split based on whether the answer to the downstream fact should change after editing. Generative Accuracy reflects whether the edited LM output agrees with the Bayesian model posterior beliefs. Probabilistic Coherence metrics are MAEs against Bayesian posterior probabilities. Logical coherence metrics reflect how the LM violates logical axioms of probability. `s1`/`s2` and `r1`/`r2` indicate the subject and relation used in the sentence, meaning `s1 r1` is the same prompt as used in model editing, while `s1 r2` is a possible downstream fact (see Fig. 4 for an example).

True/False sentences (see data example 2), with True/False values in exact proportion to the proportion of true sampled objects in the 10 noisy sentences. Then, we generate 20 logically complex sentences per subject, selecting a mixture of and/or/not sentences. The 'and' and 'or' sentences make use of other random sentences from the dataset that may be true or false. The 'not' sentences randomly uses the ground truth object for $(s, r)$ (meaning the sentence reads as: "not *s r o*" is false) or a distractor object $o_{\text{false}}$ (meaning the sentence reads as: "not *s r* $o_{\text{false}}$" is false). All of the and/or/not sentences have correct true/false labels in the pretraining dataset, to ease learning of logical connectives. Only the "*s r o*" and True/False sentences are noisy. This process is repeated until we have used 100k facts from the knowledge graph in the pretraining dataset. The result is a dataset with statistics shown in Table 6.

## A.2 Editing Benchmark Details

| Statistic | Number |
|---|---|
| True Atomic Facts | 100k |
| Atomic Sentences | 1m |
| T/F Sentences | 1m |
| and/or/not Sentences | 400k |
| Sentences | 2.4m |
| Documents | 1.26m |
| Tokens | 204m |
| Subject Entities | 47k |
| Relations | 10 |
| Object Entities | 20k |

Table 6: Pretraining dataset statistics.

Once we have pretraining data, we create the editing benchmark in two steps: (1) fit a Bayesian model to the pretraining data, (2) create test cases.

(1) Described in Sec. 6.3, fitting the Bayesian model to the pretraining dataset mostly involves counting how many times it is observed that `Arthur Green occupation architect` as opposed to `Arthur Green occupation carpenter`. These counts as used to compute the posterior predictive $p(o|s, r, \text{Upstream Property} = \varnothing)$.

Counting evidence for $p(o|s, r, \text{Upstream Property})$ is a little complicated. We wish to know how many times, e.g., someone from Harvard is an architect, someone from Yale is an architect, etc. What makes this complicated is that `educated at` is itself a noisy relation in our pretraining dataset. Whether someone went to Harvard vs. Yale is often not certain, so if that person is an architect, it is not clear *which conditional distribution* to count this evidence for. This issue is resolved by weighting observed occupation counts by frequencies of observed upstream properties. So if 80% of the time, when we see `Arthur Green`, we see `Arthur Green educated at Harvard`, and 20% fo the time we see `Arthur Green educated at Yale`, then we take their observed occupation counts $\vec{o}$ and add $.8\vec{o}$ to the evidence for $p(o|s, r = \text{occupation}, r_u = \text{educated at}, o_u = \text{Harvard})$ and $.2\vec{o}$ to the evidence for $p(o|s, r = \text{occupation}, r_u = \text{educated at}, o_u = \text{Yale})$. Recall that we aggregate this evidence across subject entities, and when determining someone's `occupation` (which is downstream of `educated at` in our hypothetical world) it does not matter what the subject entity is (all subjects share the same conditional distribution). The other complicated aspect of Bayesian inference is learning from logical sentences. True/False sentences are straightforward to interpret, but and/not/or sentences are more difficult. Here, our Bayesian model leverages the fact that these complex sentences are never noisy (always contain true atomic sentences). We compute probabilities like $p(sent_1 \text{ is true}|(sent_1 \vee sent_2) \text{ is true})$, and use these probabilities as weights for learning from the individual sentences in Bayesian model. These computations are done in `data_classes/dataset_generator.py` in our code.

(2) To create test cases for editing, we randomly sample subject entities that were seen during pretraining, then formulate test cases of the kind shown in Fig. 2. The edit requests themselves reinforce the ground truth example in the pretraining data half the time (results for fixing errors only in Table 5) and contradict the pretraining data half the time (as a typical counterfactual edit request would). The cases for measuring probabilistic coherence relate to the edit request in terms of whether they use the same or different subjects/relations. The possible downstream relation is present in the case for `s1 r2`, and we preferentially choose test cases here where we expect the downstream answer to change when updating on the requested edit (we do this 80% of the time). The logical coherence cases use another randomly selected sentence $B$, and we form True/False, and, or, and not sentences based on the edit request sentence and the sentence $B$. For all of these sentences, the Bayesian model gives posterior probabilities after updating on the edit request. For every test case, we update the Bayesian model with the edit request, and after each test case, we reset the Bayesian model so that it once again reflects the pretraining data. This updating process uses one of two weights, $n = 1000$ or $n'$ (see Sec. 6.3). The results in this paper always use $n'$.

### A.3 LM Pretraining Details

For our language model, we use the architecture from `mistralai/Mistral-7B-v0.1`. We scale the model down to 83m parameters by setting `hidden size = 512`, `intermediate size = 512*4`, `num attention heads = 8`, `num hidden layers = 12`. We train without dropout, using a batch size of 128 documents. Training for 1b tokens takes about 24 hours in an NVIDIA A6000 GPU.

### A.4 Model Editing Details

We use LoRA because it has been shown to be competitive to SOTA methods like ROME (Hua et al., 2024; Gangadhar & Stratos, 2024). In our setting, our LoRA implementation is based on `peft` (Mangrulkar et al., 2022) and is extremely similar to ROME. We perform rank-one updates to down-projection MLP layers in our language model. Since our formal language does not have paraphrases or an "is" relation, we could not leverage the paraphrase or essence regularization losses in ROME, making the methods even more similar in our setting. We use LoRA to optimize the probability of the target token conditioned on the prompt "$s\ r$", for a total of 40 gradient steps. This is sufficient to achieve 100% accuracy on `s1 r1` test cases (using the exact prompt as in the edit request).

### A.5 Logical Coherence Metrics

The probabilistic coherence metrics in Table 4 are mean absolute errors between language model probabilities and Bayesian posteriors. While we show similar idealized Bayesian probabilities for logically complex sentences in Fig. 4, the logical coherence metrics in Table 4 aim to reflect internal logical coherence of the language

model. We thus compute mean absolute errors between the left and right hand sides of the following equations reflecting basic logical axioms of probability:

$$\text{TF: } p_\theta(o|\text{``}s\ r\text{''}) = p_\theta(\text{True}|\text{``}s\ r\ o\text{''}\ \text{is})$$

$$\text{not: } p_\theta(\text{True}|\text{``}s\ r\ o\text{''}\ \text{is}) = 1 - p_\theta(\text{True}|\text{not ``}s\ r\ o\text{''}\ \text{is})$$

$$\text{or: } p_\theta(\text{True}|\text{``}s_1\ r_1\ o_1\text{''} \vee \text{``}s_2\ r_2\ o_2\text{''}\ \text{is}) =$$
$$p_\theta(\text{True}|\text{``}s_1\ r_1\ o_1\text{''}\ \text{is}) + p_\theta(\text{True}|\text{``}s_2\ r_2\ o_2\text{''}\ \text{is}) -$$
$$p_\theta(\text{True}|\text{``}s_1\ r_1\ o_1\text{''}\ \text{is}) * p_\theta(\text{True}|\text{``}s_2\ r_2\ o_2\text{''}\ \text{is})$$

$$\text{and: } p_\theta(\text{True}|\text{``}s_1\ r_1\ o_1\text{''} \wedge \text{``}s_2\ r_2\ o_2\text{''}\ \text{is}) =$$
$$p_\theta(\text{True}|\text{``}s_1\ r_1\ o_1\text{''}\ \text{is}) * p_\theta(\text{True}|\text{``}s_2\ r_2\ o_2\text{''}\ \text{is})$$

We note that our model easily learns that $p_\theta(\text{True}|\text{``}s\ r\ o\text{''}\ \text{is})$ and $p_\theta(\text{False}|\text{``}s\ r\ o\text{''}\ \text{is})$ should sum to 1, so we can compute all of the above metrics using "True" as the target probability.

## B  Additional Results

| | Generative Accuracy ↑ | | | | Probabilistic Coherence ↓ | | | | Logical Coherence ↓ | | | |
|---|---|---|---|---|---|---|---|---|---|---|---|---|
| | s1 r1 | s1 r2 | s2 r1 | s2 r2 | s1 r1 | s1 r2 | s2 r1 | s2 r2 | TF | neg. | and | or |
| *Embeddings* | | | | | | | | | | | | |
| Pre-edit | 0.96 | 0.93 | 0.92 | 0.92 | 0.40 | 0.22 | 0.34 | 0.21 | 0.40 | 0.22 | 0.34 | 0.21 |
| Post-edit | 1.00 | 0.29 | 0.75 | 0.91 | 0.51 | 0.28 | 0.34 | 0.24 | 0.51 | 0.28 | 0.34 | 0.24 |
| Δ | +.04 | −.64 | −.17 | −.01 | +.11 | +.06 | +.00 | +.03 | +.11 | +.06 | +.00 | +.03 |
| *LORA-all* | | | | | | | | | | | | |
| Pre-edit | 0.96 | 0.93 | 0.92 | 0.92 | 0.40 | 0.22 | 0.34 | 0.21 | 0.40 | 0.22 | 0.34 | 0.21 |
| Post-edit | 0.99 | 0.76 | 0.91 | 0.92 | 0.52 | 0.22 | 0.34 | 0.21 | 0.52 | 0.22 | 0.34 | 0.21 |
| Δ | +.03 | −.17 | −.01 | −.00 | +.12 | +.00 | +.00 | +.00 | +.12 | +.00 | +.00 | +.00 |
| *SGD* | | | | | | | | | | | | |
| Pre-edit | 0.96 | 0.93 | 0.92 | 0.92 | 0.40 | 0.22 | 0.34 | 0.21 | 0.40 | 0.22 | 0.34 | 0.21 |
| Post-edit | 1.00 | 0.76 | 0.91 | 0.92 | 0.52 | 0.23 | 0.34 | 0.21 | 0.52 | 0.23 | 0.34 | 0.21 |
| Δ | +.04 | −.17 | −.01 | −.00 | +.12 | +.00 | +.00 | +.00 | +.12 | +.00 | +.00 | +.00 |

Table 7: Model editing results **with different editing methods**. We use embedding-only finetuning (embeddings), LORA on all attention and MLP weights, and whole-model finetuning with SGD. Test data is split based on whether the answer to the downstream fact should change after editing. Generative Accuracy reflects whether the edited LM output agrees with the Bayesian model posterior beliefs. Probabilistic Coherence metrics are MAEs against Bayesian posterior probabilities. Logical coherence metrics reflect how the LM violates logical axioms of probability. s1/s2 and r1/r2 indicate the subject and relation used in the sentence, meaning s1 r1 is the same prompt as used in model editing, while s1 r2 is a possible downstream fact.

We conduct additional experiments with three editing methods: (1) embedding-only finetuning, (2) whole-model finetuning, and (3) LORA on all attention and MLP weights. The (1) method should represent a weak baseline, as finetuning embeddings only should warp the meaning of subject and relation embeddings. Meanwhile, (2) and (3) should be stronger.

We show the results in Table 7 for all edit requests. The accuracy columns show agreement between the edited LM and labels given by the Bayesian model. The last four columns are logical coherence metrics, given as the absolute difference between the LM probability and what a logically consistent model should output (see paper for more detail).

Indeed, when we compare results between the methods, we see a reasonable gradation in performance between methods. Finetuning embeddings overgeneralizes to test cases with the same subject or relation, and does more damage to the model's logical consistency. Meanwhile, whole model finetuning and LORA achieve similar performance, with a slight tradeoff between editing the output for the edit request (s1/r1) and overgeneralizing to other cases whose answers should not change (s2/r1 and s2/r2). All methods perform extremely poorly at generalizing to downstream facts whose answer may change following the edit request (s1/r2), a finding unique to our benchmark.

| | Generative Accuracy | Coherence Error ↓ | |
|---|---|---|---|
| | s1 r1 | TF | neg. |
| *All Requested Changes* | | | |
| Pre-edit | 0.96 | 0.40 | 0.22 |
| Post-edit | 1.00 | 0.52 | 0.23 |
| Δ | +.04 | +.12 | +.01 |
| *Initially Logically Coherent Data* | | | |
| Pre-edit | 0.99 | 0.02 | 0.03 |
| Post-edit | 1.00 | 0.75 | 0.05 |
| Δ | +.01 | +.73 | +.02 |

Table 8: Comparison of editing performance on all requested changes versus points with initially logically coherent probability assignments across T/F and negated versions of facts. Even for points with initial logical coherence, weight editing can destroy logical coherence across T/F versions of a claim ("*X* is true") after updating the model.

