# OpenReview forum: "Fundamental Problems With Model Editing: How Should Rational Belief Revision Work in LLMs?"
_TMLR — Accepted by TMLR_

### Review · Reviewer_fxL7 · 2024-08-07

**Summary Of Contributions:**

The paper discusses the challenges of model editing in LLM. It first defines 12 different challenges that fall into three categories: defining the model editing problem, benchmark characteristics, how to update the LLM belief (if this is possible). Then, it presents a possible benchmark.

**Audience:**

Yes

**Broader Impact Concerns:**

I think the Broader Impact section is fair enough. I do not have any concerns.

**Claims And Evidence:**

Yes

**Requested Changes:**

Perhaps another possible problem is missing: could the addition of information lead the model to forget other, not necessarily conflicting, information?
This does not seem to me to be a problem considered by any of the others already listed, but I could be wrong. I would like to know what the authors think.

Minor issue:
On page 8 and on page 10, the first row of sections 5 and 6 should refer to Table 1

**Strengths And Weaknesses:**

The paper discusses the challenges of model editing in LLM. It first defines 12 different challenges that fall into three categories: defining the model editing problem, benchmark characteristics, and how to update the LLM belief (if this is possible). Then, it presents a possible benchmark.

The paper is interesting for its key idea. It wants to discuss many different problems that should be solved to perform model editing. It then leaves the solution of each of these problems to future work.
However, in my opinion, the paper lacks a proper discussion of possible approaches. Some of the proposed problems have already been addressed by other authors, as evidenced by the references given, but they are usually only mentioned instead of being analysed properly (what strengths and weaknesses should be exploited or avoided, for example).

In the end, the document presents a possible formal test case. Although this is an important strength of the paper, I believe that a more thorough discussion could help the reader gain a deeper insight into the results obtained and what the benchmark aims to test against the problems listed and discussed above.
In this section, Table 4 should be discussed in detail, along with the meaning of consistency. For example, if I have not missed anything, the arrows say that a lower level is better, but why?

---

> ### Author Response · Authors · 2024-09-13
> **Reply to Reviewer fxL7**
>
> > Some of the proposed problems have already been addressed by other authors, as evidenced by the references given, but they are usually only mentioned instead of being analysed properly (what strengths and weaknesses should be exploited or avoided, for example)
>
> We understand that the concern is roughly that we are raising many challenges with the problem but not necessarily offering a path forward with explicit suggestions. We have correspondingly updated the paper pdf with additional concrete suggestions for future work in Secs. 3.4, 5.1, and 5.2. Although, respectfully, we point out that we often do offer specific suggestions in other sections. For example: (1) in Sec. 4.2 we state that “Future benchmarks will need to carefully select claims that are less vague and ambiguous.”; (2) in Sec. 4.3 we point out that “It will not be sufficient to gather large collections of generic factual claims about the world that LLMs already know most of”; and (3) in Sec 5.3 we recommend further work on how LLMs metalearn a notion of evidentiary value of text in order to design better model editing methods.
>
> Moreover, some of the challenges will not be settled in one paper. Philosophical problems such as the problems of background beliefs (Sec. 3.1) and many possible worlds (Sec. 3.2) are long standing challenges, and our main goal with these sections is to alert the computer science community to how they apply to model editing, rather than make headway on possible solutions there. In addition, we hope that our proposed benchmark itself provides some guidance on where to go next, since we mitigate issues due to background beliefs and possible worlds by using a formal language and comparing against a Bayesian model (see also new Sec. 6.1 on how the benchmark mitigates issues from the 12 challenges).
>
> This all said, we are happy to try to be more explicit about future practices sections that do currently end with less concrete suggestions. As mentioned above, we have updated the endings of Secs. 3.4, 5.1, and 5.2 to try to be more direct about what we hope to see future work focus on.
>
> > Table 4 should be discussed in detail, along with the meaning of consistency
>
> We have expanded the discussion of Table 4 to try to more fully walk readers through the notation and results! See the new text in red. We have also tried to line up the text with the figures/tables better to make it easier to refer back and forth.
>
> > Perhaps another possible problem is missing: could the addition of information lead the model to forget other, not necessarily conflicting, information?
>
> Very interesting! Indeed, we could see this being another problem, possibly falling under the header of “Challenges With Assuming LLMs Have Editable Beliefs.” We think this is possible if the model has limited memory capacity. Past work has investigated direct knowledge storage in model weights and scaling capacity [1, 2]. There should be theoretical limits on model storage that could lead to natural forgetting of old facts as more and more new facts are adopted, particularly when edits focus on localized components of the network like single matrices or subspaces of matrix weights. There is new ground to be explored in the relationship between localization of knowledge in models and understanding total memory capacity.
>
> What do you think about this kind of case for natural forgetting during model editing? We could build this out as a 13th problem in the paper, after digging more into empirical and theoretical work on the problem from the CS community, computational neuroscience [3], etc.
>
> [1] How Much Knowledge Can You Pack Into the Parameters of a Language Model?
> [2] Quantifying Memorization Across Neural Language Models
> [3] Universal Hopfield Networks: A General Framework for Single-Shot Associative Memory Models
>
>
> > On page 8 and on page 10, the first row of sections 5 and 6 should refer to Table 1
>
> Thank you! Fixed.

---

> > ### Comment · Reviewer_fxL7 · 2024-09-16
> >
> > On the future work parts, I think they would be enough given the nature of this paper.
> >
> > On the discussion about the possible 13th problem, the problem I suggested probably partly falls in the “Challenges With Assuming LLMs Have Editable Beliefs.” However, in this case, the name should be updated to better reflect this new part as well. As it is now, it makes me think more about not forgetting but simply changing the model's mind about something (e.g., the model knows that all birds fly but after knowing about penguins it changes its mind by saying that most birds fly).
> >
> > Probably, also considering your answer, it would be better to add a new problem to keep the two concepts separate.

---

### Review · Reviewer_ZXqR · 2024-08-30

**Summary Of Contributions:**

This work summarizes key challenges, i.e., open problems, in model editing for large language models and proposes a semi-synthetic dataset to benchmark the performance of model editing. The problems are summarized in terms of three view points, task settings, benchmark dataset construction and future impacts. The benchmark dataset is constructed from Wikidata by focusing on subset of relation pairs, e.g., occupation and education, which are related with each other, i.e.g, downstream relation and upstream relation. The synthetically created dataset is used to train an LLM to evaluate the performance of model editing by altering particular relations and by measuring the coherence of relations. Experiments show that simply editing a tuple does not reflect the changes in downstream relations, demonstrating the limitation of a naive fine-tuning based editing.

**Audience:**

Yes

**Broader Impact Concerns:**

No concerns.

**Claims And Evidence:**

Yes

**Requested Changes:**

I'd like to find more discussion on how the proposed dataset is addressing the open problems mentioned in section 3 through 5.

**Strengths And Weaknesses:**

Strengths

- It is a good survey on the challenges for model editing, summarizing into three interesting view points with a lot of details. I feel the summary could be a nice contribution to the field.

- The semi-synthetic dataset is another good contribution in that it is well designed to control the setting by sampling tuples from Wikidata, and the metrics for coherence is a sound method based on Bayesian posteriors. I think the experiments are solid to demonstrate the current model editing capabilities.

Weaknesses

- Although the semi-synthetic dataset is a good contribution, it is not immediately clear what open problems the dataset is addressing in the summary of open problems mentioned in section 3 through 5.

- This work is somehow limited in that it is focusing on task settings, and not discussing potential methods for model editing besides fine-tuning with LoRA. Although I understand that methodologies are out of the focus of this work, it would be good to add some discussion on how researchers challenges the problem.

---

> ### Author Response · Authors · 2024-09-13
> **Reply to Reviewer ZXqR**
>
> > it is not immediately clear what open problems the dataset is addressing in the summary of open problems mentioned in section 3 through 5
>
> Thanks for flagging this, as we currently describe this motivation in only 1-2 sentences. We have updated the pdf to include another small subsection to more thoroughly connect our benchmark to the previously described challenges. See the new Sec. 6.1.
>
> > This work is somehow limited in…not discussing potential methods for model editing besides fine-tuning with LoRA
>
> We have added results with more editing methods, including finetuning embeddings, whole-model finetuning, and different variations of LORA. See the response to Reviewer rqA9 for a table with results.
>
> Besides that, we would justify our focus on weight-based model editing by pointing to how this is a predominant paradigm in the literature that has seen a huge influx of new papers (see 17 papers cited in paper Introduction). Please let us know if this does not fully address your concern.

---

### Review · Reviewer_rqA9 · 2024-08-31

**Summary Of Contributions:**

This work is a constructive criticism of model editing, or what the authors call "belief revision": meaningfully revising knowlege captured by a generative model. The authors categorize possible issues with belief revision, including questioning whether generative models have beliefs to begin with. They show examples and explanations of these categories. This includes highlighting issues with current benchmarks, e.g. having ambiguous or incorrect ground truth labels.

To address some of the concerns and issues, this work introduces a dataset that can be used to test model editing performance. This benchmark task involves fitting a Bayesian model to the output of a generative model, and measuring whether the update in probabilities matches how a rational agent would update its beliefs. The authors' experiments indicate that - assuming LLMs can indeed hold beliefs - model editing through LoRA-based finetuning can easily result in inconsistent beliefs.

**Audience:**

Yes

**Broader Impact Concerns:**

Agreed with current impact statement.

**Claims And Evidence:**

Yes

**Requested Changes:**

- For section 6: is there a case where the model belief corresponds to the rational pre-update credence? If a generative model cannot get the belief for "A is true" and "not A is true" to complement each other, then it may not provide much signal for an editing method either.
- Please provide a bit more information for the columns in Fig 4., it took me longer than I'd like to admit to fully understand how the "same r", "diff s" labels corresponded to the example sentence. Since this figure presents a qualitative editing result that shows the core issue, I think you can afford to spend more text on it, similar to table 4.
- In section 6, the authors try one editing method (LoRA based finetuning), and at the end of section 6 highlight in bold that 'model edits totally fail to generalize'. Given that this sentence is in bold, it would be more accurate to say that this specific editing method fails, lest readers believe this is a general claim about model editing methods.

**Strengths And Weaknesses:**

Strengths:
- The categorization proposed here is clearly explained. I believe this is a useful attempt that will help future discussion on model belief updating. Perhaps categories 5.1 or 5.2 are bordering a bit on the philosophical, or contain anthropomorphisms (discussing the *aims* of LLMs), which makes them less practical. But the questions they posit are still useful, in my opinion.
- The proposed model editing dataset is quite large, making it well-suited for (small) large language model experiments - even if the sentence structure is rigid.
- The benchmark task involves relations that do not have 1-1 mappings per se, which makes it well-suited for evaluating updates to probabilities. It also seems more realistic than the synthetic corpus from Betz and Richardson.
- I believe benchmarking model editing, and trying to quantify "beliefs" held by language model weights is useful and interesting for the TMLR audience.

Weaknesses:
- Although the benchmark task is a useful test setting for model editing methods, it is much simpler than real world cases. The number of relations is small (10), and the structure of the sentences is rigid. If an editing method does well on this benchmark task, it does not necessarily mean it will do well on real world unstructured text. In other words, the task may be too synthetic to be practical.
- Only one editing method is tried here, it would be stronger to try at least two: one weak baseline, and one method considered by the community to be a strong editing method. If the benchmark results show a meaningful difference between these two methods, it gives me additional confidence that the benchmark task can serve as proxy for real world model editing tasks.

---

> ### Author Response · Authors · 2024-09-13
> **Reply to Reviewer rqA9 (part 1)**
>
> > The number of relations is small (10)
>
> It’s true that only 10 relations is fairly small. We confirmed that we can generate a pretraining corpus with 100 relations by reading up to 7m facts from Wikidata. We will release this corpus along with the code for the work.
>
> > If an editing method does well on this benchmark task, it does not necessarily mean it will do well on real world unstructured text
>
> We appreciate this concern and agree with it in principle, but ultimately we believe that our synthetic dataset is a strong and unique contribution toward having sound model editing benchmarks because even popular “real world” benchmarks have suffered from generalizability issues in the past, possibly due to problems like we point out in the 12 challenges in our paper. For instance, methods that do well on CounterFact often do not generalize to other simple evaluations [1]. We hope that since our work aims to clarify how model editing benchmarks can be underspecified, it will provide a clearer basis for more naturalistic datasets in the future.
>
> > Only one editing method is tried here, it would be stronger to try at least two…If the benchmark results show a meaningful difference between these two methods, it gives me additional confidence that the benchmark task can serve as proxy for real world model editing tasks.
>
> This point is well taken and we’ve conducted more experiments to include three editing methods: (1) embedding-only finetuning, (2) whole-model finetuning, and (3) LORA on all attention and MLP weights. The (1) method should represent a weak baseline, as finetuning embeddings only should warp the meaning of subject and relation embeddings. Meanwhile, (2) and (3) should be stronger. We show the results below for all edit requests. The accuracy columns show agreement between the edited LM and labels given by the Bayesian model. The last four columns are logical coherence metrics, given as the absolute difference between the LM probability and what a logically consistent model should output (see paper for more detail).
>
> | method     | condition | same_s_same_r_acc | same_s_diff_r_acc | diff_s_same_r_acc | diff_s_diff_r_acc | TF_coherence | negation | multiplication | disjunction |
> |------------|-----------|-------------------|-------------------|-------------------|-------------------|--------------|----------|----------------|-------------|
> | embeddings | pre-edit  | 0.964             | 0.9262            | 0.9219            | 0.9249            | 0.3996       | 0.2236   | 0.3373         | 0.2109      |
> | embeddings | post-edit | 1                 | 0.286             | 0.7503            | 0.9068            | 0.5096       | 0.2789   | 0.3412         | 0.2422      |
> | embeddings | delta     | **0.036**         | **-0.6402**       | **-0.1716**       | **-0.0181**       | **0.11**     | **0.0553** | **0.0039**     | **0.0313**  |
> | LORA-all   | pre-edit  | 0.964             | 0.9262            | 0.9219            | 0.9249            | 0.3996       | 0.2236   | 0.3373         | 0.2109      |
> | LORA-all   | post-edit | 0.9871            | 0.7629            | 0.9148            | 0.9211            | 0.5173       | 0.2246   | 0.3374         | 0.2116      |
> | LORA-all   | delta     | **0.0231**        | **-0.1633**       | **-0.0071**       | **-0.0038**       | **0.1177**   | **0.001** | **1.00E-04**   | **7.00E-04** |
> | SGD        | pre-edit  | 0.964             | 0.9262            | 0.9219            | 0.9249            | 0.3996       | 0.2236   | 0.3373         | 0.2109      |
> | SGD        | post-edit | 1                 | 0.7626            | 0.9138            | 0.9191            | 0.5192       | 0.225    | 0.3376         | 0.212       |
> | SGD        | delta     | **0.036**         | **-0.1636**       | **-0.0081**       | **-0.0058**       | **0.1196**   | **0.0014** | **3.00E-04**   | **0.0011**  |
>
> Indeed, when we compare results between the methods, we see a reasonable gradation in performance between methods. Finetuning embeddings overgeneralizes to test cases with the same subject or relation, and does more damage to the model's logical consistency. Meanwhile, whole model finetuning and LORA achieve similar performance, with a slight tradeoff between editing the output for the edit request (same_same_r_acc) and overgeneralizing to other cases whose answers should not change (diff_s_same_r_acc and diff_s_diff_r_acc). All methods perform extremely poorly at generalizing to downstream facts whose answer may change following the edit request (same_s_diff_r_acc), a finding unique to our benchmark.
>
> [1] Detecting Edit Failures In Large Language Models: An Improved Specificity Benchmark

---

> > ### Author Response · Authors · 2024-09-13
> > **Reply to Reviewer rqA9 (part 2)**
> >
> > > is there a case where the model belief corresponds to the rational pre-update credence?
> >
> > Good question! We are able to filter to a subset of test points for which the model does achieve rational pre-update credences and confirm that model editing performance is similarly poor on this subset. Specifically, we filter to data where logical coherence is respected within a small margin of error, for True/False and negation coherence, meaning p(A) = p(A is true) and p(A) = 1 - p(not A) with less than .05 absolute difference in these quantities. On this subset of test cases, we find that model edits still fail to generalize to entailed data and actively harm logical coherence of models. For instance, while models may initially obtain good coherence for p(A) = P(A is true), updates often totally fail to generalize to True/False propositions, leading to the absolute difference in these quantities being 0.73 on average (TF coherence error delta for Initially Logically Coherent Data).
> >
> > | Data | condition | same_s_same_r_acc ↑ | TF coherence error ↓ | negation coherence error ↓ |
> > |------|-----------|----------------------|-----------------|------------|
> > | All Requested Changes | pre-edit  | 0.96                 | 0.4             | 0.22       |
> > | | post-edit | 1                    | 0.52            | 0.23       |
> > | | delta     | 0.04                 | 0.12            | 0.01       |
> > | Initially Logically Coherent Data | pre-edit  | 0.99                 | 0.02            | 0.03       |
> > | | post-edit | 1                    | 0.75            | 0.05       |
> > | | delta     | 0.01                 | 0.73            | 0.02       |
> >
> > More broadly, we know that LLMs still fail to maintain coherent credences in propositions in many cases. This is a limitation of the models and not necessarily model editing methods, as you point out. While we consider it out of scope for this work to improve LLM logical consistency, this potential shortcoming of the edited LLM is something that our benchmark should help indicate by including evaluations for logical coherence – these metrics are not included in any past benchmarks that we know of.
> >
> > > Please provide a bit more information for the columns in Fig 4
> >
> > We have updated the pdf to include more description in the main body to help readers understand the same/diff s/r notation in Fig. 4 and Table 4! We have also tried to line up the text with the figures/tables better to make it easier to refer back and forth.
> >
> > >  'model edits totally fail to generalize'...Given that this sentence is in bold, it would be more accurate to say that this specific editing method fails
> >
> > Thanks for pointing this out. With our new results on 3 editing methods, we hope that this statement is now well substantiated, but will adjust it to claim that “_Popular_ editing methods totally fail to generalize”

---

### Author Response · Authors · 2024-09-13
**General response**

Thanks to all the reviewers for the positive comments and thoughtful feedback! We are happy to see that reviewers believe the paper to be “useful and interesting for the TMLR audience” (rqA9), with “challenges for model editing…[that] could be a nice contribution to the field” (ZXqR) and a benchmark that is “another good contribution” (ZXqR) and “an important strength of the paper” (fxL7). We thank the reviewers for their time and attention invested.

We plan on replying to each reviewer’s questions individually below, and we have updated the submission pdf to reflect updates to the text of the paper in response to some writing suggestions as well (new text in red), so you can reference that as needed when reading individual replies.

---

### Decision · Action_Editor_VfCJ · 2024-10-08

**Recommendation:** Accept with minor revision

**Comment:**

All reviewers appreciated the paper, the interaction with the authors and the revisions made. The reviewers hope that minor concerns not fully addressed in the revision will still be addressed in the final minor revision.

Quoted from Reviewer rqA9:

I think the benchmark task is a useful attempt at measuring editing performance. There are still some minor concerns:

* authors added a section connecting issues to benchmark properties, but the connection is a bit superficial for some,
* the task is a bit synthetic (while they did update the number of relations, making it more realistic)
* the authors only try out three variations of the same simple editing method

**Audience:**

Yes.

**Claims And Evidence:**

Yes.